# Initialization is Half the Battle: Generating Diverse Images from a Guidance Potential Posterior

Xiang Li [1]  Dianbo Liu [1]  Kenji Kawaguchi [1]

## Abstract

Despite the remarkable fidelity of generative models, they frequently suffer from mode collapse. Existing strategies for enhancing diversity predominantly focus on intervening during the generation trajectory. We identify a critical oversight that the standard Gaussian initialization often causes trajectories to collapse into dominant modes because it is agnostic to the guidance potential landscape. In this work, we formulate selecting the initial noise from a *guidance potential posterior*, which effectively re-weights the prior towards diversity-rich regions. To sample from this distribution efficiently, we introduce *Diversity-inducing Initialization* (`DivIn`), which leverages Langevin dynamics to actively navigate the initialization landscape, steering initial noise away from collapsing regions while anchoring them to the valid data manifold. Our method serves as an inference-time diversity enhancement compatible with both diffusion and flow matching models. Extensive experiments show that `DivIn` exhibits a superior performance in both class-to-image and text-to-image scenarios. Furthermore, we highlight that as `DivIn` is orthogonal to trajectory-based methods, combining them significantly expands the diversity-quality Pareto frontier beyond what either achieves in isolation.

## 1. Introduction

Generative diffusion models and flow matching models (Ho et al., 2020; Rombach et al., 2022; Lipman et al., 2023; Esser et al., 2024) have achieved remarkable success in synthesizing high-fidelity images from text prompts. However, despite their stochastic formulation, these models frequently suffer from *mode collapse*. Even with varying random seeds,

[1]National University of Singapore. Correspondence to: Xiang Li <xiangli@comp.nus.edu.sg>.

*Proceedings of the 43rd International Conference on Machine Learning*, Seoul, South Korea. PMLR 306, 2026. Copyright 2026 by the author(s).

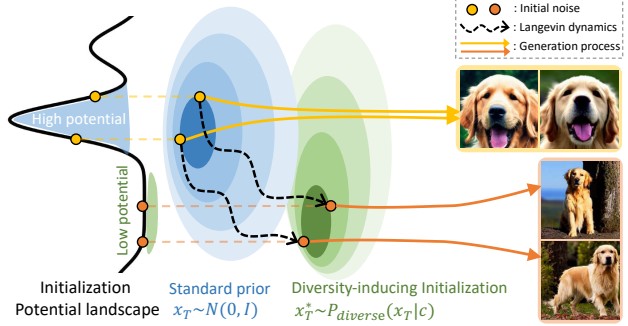

*Figure 1.* Standard prior (blue) often lands in sharp, high potential hills that attract trajectories into a single dominant mode (mode collapse). Without changing the generation process, our proposed `DivIn` (green) leverages Langevin dynamics to sample from flat, low-potential basins as a diversity posterior, while staying anchored to the valid Gaussian distribution to enhance the generation diversity-quality trade-off.

generated samples often converge to stereotypic outputs, fundamentally limiting the models' creative potential (Ho & Salimans, 2022; Shipard et al., 2023; Morshed & Boddeti, 2025), and lead to exact memorization of training data in extreme cases (Carlini et al., 2023; Somepalli et al., 2023; Wen et al., 2024).

Existing training-free methods to enhance generative diversity mainly focus on intervening in the sampling process, for example, by manipulating guidance scales (Sehwag et al., 2022; Sadat et al., 2024; Kynkäänniemi et al., 2024), or actively repelling trajectories from one another (Corso et al., 2024; Kirchhof et al., 2025; Morshed & Boddeti, 2025). While effective, these methods largely neglect the starting point, assuming that the standard isotropic Gaussian initialization is sufficient for discovering diverse modes.

In this work, we reveal that initialization is *the other half of the battle*. Motivated by recent findings linking local score landscape sharpness to memorization (Jeon et al., 2025), we generalize this geometric perspective to the broader challenge of mode collapse. Geometrically, the regions where the conditional guidance potential is high induce high volume contraction rates in the generative flow. Since the standard Gaussian initialization is oblivious to the landscape of the conditional guidance, distinct stochastic trajectories initialized in these regions are often attracted into a sin-

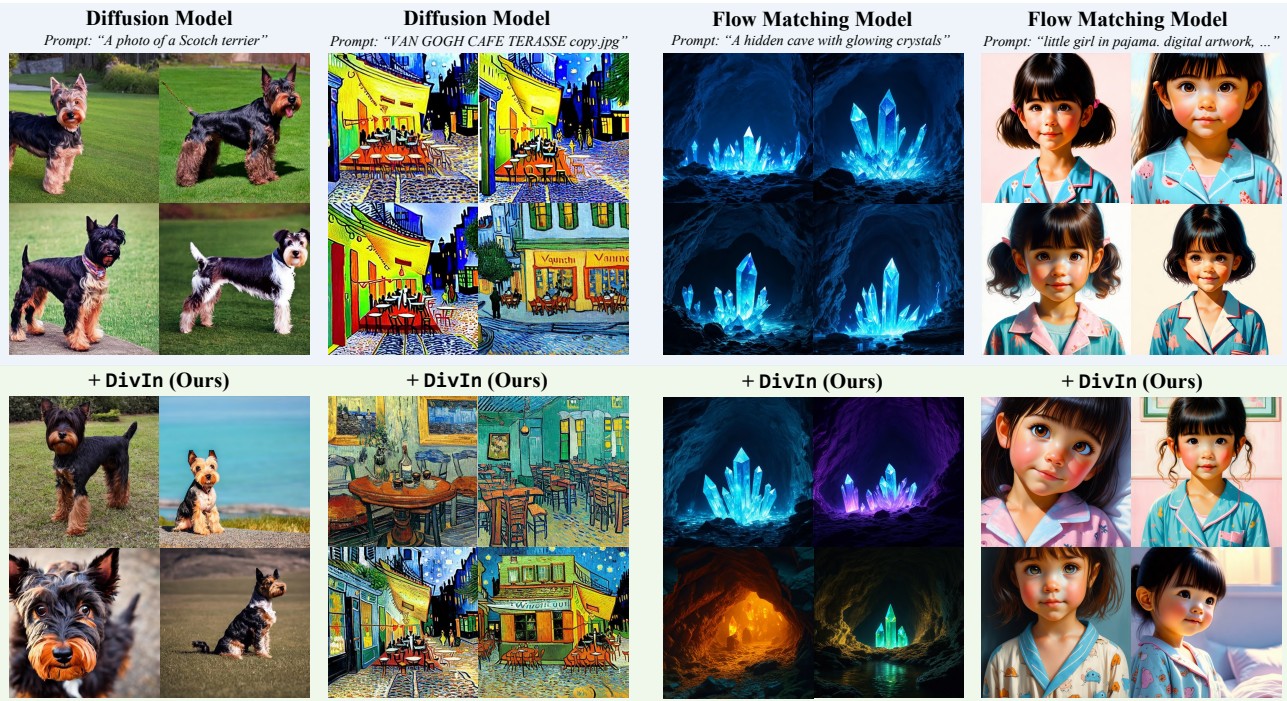

**Figure 2.** Qualitative comparison on various prompts and different architectures. By sampling initial noise from regions with low guidance potential, `DivIn` (bottom row) discovers more modes and generates diverse images.

gle dominant mode. In contrast, sampling initial noises from low-potential basins allows trajectories to naturally diverge, leading to the discovery of diverse modes. However, we observe that deterministic seed optimization methods are inherently mode-seeking and prior-breaking, and thus inevitably collapse the initialization distribution and undermine generative diversity (Jeon et al., 2025).

To address this limitation, we formulate the optimal initialization not as a fixed prior, but as a *guidance potential posterior* that re-weights the standard prior towards regions where diversity originates. We introduce `DivIn` (*Diversity-inducing Initialization*) that leverages Langevin dynamics to approximate this posterior. As illustrated in Figure 1, `DivIn` utilizes a computationally efficient proxy for the guidance potential to actively navigate the initialization landscape. Unlike deterministic optimization that may drift off-manifold, our formulation maintains manifold validity while enabling robust exploration of diverse modes.

Our method serves as a universal, plug-and-play strategy compatible with both standard diffusion and modern flow matching models, as shown in Figure 2. Extensive experiments demonstrate that `DivIn` significantly improves diversity while maintaining image quality in both class-to-image and text-to-image tasks. Moreover, we highlight that as initialization is independent of the subsequent generation process, our method provides a distinct source of diversity not captured by trajectory interventions. Combining

`DivIn` with existing sampling mechanisms further expands the Pareto frontier of the diversity-quality trade-off.

## 2. Related Work

**Diversity in Generative Models** It remains a significant challenge for modern generative models to trade off between sample quality and diversity (Ho & Salimans, 2022; Shipard et al., 2023; Dombrowski et al., 2025). Existing approaches to mitigate this issue generally fall into two categories: training-time and inference-time enhancements. Training-time methods necessitate retraining or fine-tuning the model parameters to encourage diverse mode coverage, utilizing techniques such as reinforcement learning objectives (Miao et al., 2024), entropy maximization (De Santi et al., 2025), or ambient diffusion (Shah et al., 2025). Training-free methods intervene in the sampling trajectory during the inference phase. These include modified sampling strategies to target low-density regions (Sehwag et al., 2022; Sadat et al., 2024; Kynkäänniemi et al., 2024) or gradient-based guidance that actively pushes samples apart (Corso et al., 2024; Kirchhof et al., 2025; Morshed & Boddeti, 2025). In contrast, as a training-free method, `DivIn` focus exclusively on modifying the initial noise distribution $x_T$, leaving the generative process untouched. Consequently, our approach is orthogonal to existing sampling-based diversity methods and can be seamlessly combined with them to further improve the diversity-quality trade-off.

**Impact of the Initial Noise** Recent research has highlighted the structural significance of the initial noise in diffusion models (Samuel et al., 2023). It has been observed that the initial noise correlates with specific regions in the generated output (Mao et al., 2023), and optimizing it can improve text-image alignment (Guo et al., 2024). Furthermore, specific seeds can be discovered to generate rare concepts (Samuel et al., 2024), achieve high fidelity (Xu et al., 2025), or mitigate memorization (Jeon et al., 2025; Han et al., 2025). However, existing noise optimization methods typically rely on the entire generative chain to minimize a loss defined on the final image or deterministic optimization to find a single optimal seed. Our work fundamentally extends the understanding of the initialization potential landscape to generation diversity. `DivIn` minimizes potential at the initial sampling step with minimal computation cost and employs Langevin dynamics to preserve manifold volume.

## 3. Preliminaries

**Diffusion Models** Diffusion models (Sohl-Dickstein et al., 2015; Ho et al., 2020; Song et al., 2021b) generate data by reversing a stochastic process that gradually maps the data distribution $p_{data}(\mathbf{x})$ to an isotropic Gaussian prior $\mathcal{N}(0, \mathbf{I})$. The forward process is modeled as a Stochastic Differential Equation (SDE) that progressively injects noise into the clean data $\mathbf{x}_0$ over time $t \in [0, T]$: $d\mathbf{x} = \mathbf{f}(\mathbf{x}, t)dt + g(t)d\mathbf{w}$, where $\mathbf{w}$ is a standard Brownian motion. The Variance Preserving (VP) formulation is defined as $\mathbf{x}_t = \sqrt{\overline{\alpha}_t}\mathbf{x}_0 + \sqrt{1 - \overline{\alpha}_t}\epsilon$, where $\epsilon \sim \mathcal{N}(0, \mathbf{I})$. Sampling is performed by solving the reverse-time SDE:

$$d\mathbf{x} = [\mathbf{f}(\mathbf{x}, t) - g^2(t)\nabla_{\mathbf{x}} \log p_t(\mathbf{x})]dt + g(t)d\bar{\mathbf{w}}. \quad (1)$$

In practice, a neural network $\epsilon_\theta(\mathbf{x}_t, t, c)$ is trained to approximate the noise, which is proportional to the score function $\nabla_{\mathbf{x}} \log p_t(\mathbf{x})$. The clean data estimate $\hat{\mathbf{x}}_0$ at any timestep can be recovered via Tweedie's formula:

$$\hat{\mathbf{x}}_0(\mathbf{x}_t) = \frac{\mathbf{x}_t - \sqrt{1 - \overline{\alpha}_t}\epsilon_\theta(\mathbf{x}_t)}{\sqrt{\overline{\alpha}_t}}. \quad (2)$$

**Flow Matching Models** Flow Matching and Rectified Flow (Lipman et al., 2023; Esser et al., 2024) offer an alternative continuous-time generative framework that connects data and noise distributions via an Ordinary Differential Equation (ODE). The framework defines a probability flow ODE $\frac{d\mathbf{x}}{dt} = \mathbf{v}_\theta(\mathbf{x}, t, c)$, where the vector field $\mathbf{v}_\theta$ learns to reverse the velocity of the interpolation path $\mathbf{x}_t = (1 - t)\mathbf{x}_0 + t\mathbf{x}_1$, with $\mathbf{x}_1 \sim \mathcal{N}(0, \mathbf{I})$ serving as the noise prior. The linear structure of the flow allows for a direct geometric estimation of the clean data $\hat{\mathbf{x}}_0$ from any state $\mathbf{x}_t$ by projecting along the flow direction:

$$\hat{\mathbf{x}}_0(\mathbf{x}_t) = \mathbf{x}_t - t \cdot \mathbf{v}_\theta(\mathbf{x}_t, t). \quad (3)$$

## 4. Method

In this section, we present a geometry-aware initialization framework designed to unlock the latent diversity of diffusion models. We first analyze the geometric bottlenecks in the initial noise landscape that lead to mode collapse (Section 4.1). We then formulate diversity as a sampling problem from a modified diversity-weighted posterior (Section 4.2). Finally, we derive a Langevin dynamics algorithm to efficiently sample from this distribution while preserving the validity of the generative manifold (Section 4.3).

### 4.1. Motivation

Standard diffusion sampling operates under the assumption that the isotropic Gaussian prior $\mathbf{x}_T \sim \mathcal{N}(0, \mathbf{I})$ provides a sufficient and unbiased seed for generation. However, we present that this assumption fails to account for the *non-uniform local geometry* of the initialization landscape. While recent work has linked initialization landscape to memorization (Jeon et al., 2025; Wen et al., 2024; Han et al., 2025), we generalize this geometric perspective to the broader challenge of generative diversity.

We hypothesize that mode collapse is driven by the landscape of the conditional guidance at initialization. Geometrically, dominant modes often form deep basins of attraction characterized by high local sharpness. As we formally derive in Appendix A.2 (Theorem A.3), regions of high curvature induce a rapid rate of probability volume contraction during the reverse process. As illustrated in Figure 3, this forces distinct stochastic trajectories to merge into stereotypic outputs. Conversely, biasing the initialization toward the regions with a less contractive force preserves diversity.

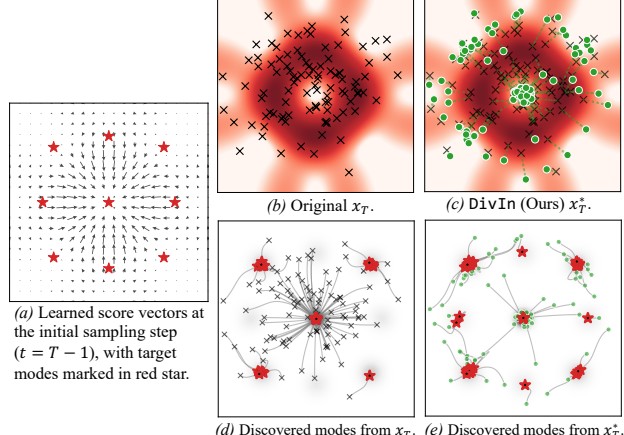

*(b) Original $x_T$.*     *(c) `DivIn` (Ours) $x_T^*$.*

*(a) Learned score vectors at the initial sampling step ($t = T - 1$), with target modes marked in red star.*

*(d) Discovered modes from $x_T$.*     *(e) Discovered modes from $x_T^*$.*

*Figure 3.* Comparison of mode discovery on a 2D toy distribution with 9 modes. *(b)* Standard Gaussian initialization (black "x") concentrates samples in the high-potential region (dark red) driven by the central dominant mode, leading to *(d)* mode collapse where only 5 modes are recovered. *(c)* Our proposed initialization ($x_T^*$, green "o") disperses samples across the low-potential landscape, successfully recovering *(e)* all 9 modes.

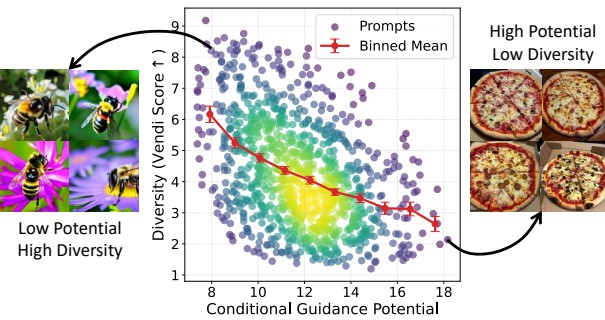

*Figure 4.* We analyze the initialization potential against generative diversity. We sample 10 independent initial noises for each of the 1000 distinct prompts and calculate the Vendi score on a kernel of 10 and average potential across the 10 latents for each prompt. The binned mean (red line) reveals a strong negative correlation that initial noises with high guidance potential lead to mode collapse, while those with low potential preserve diversity.

To empirically validate this hypothesis, we analyze the correlation between geometric potential and generative diversity across 1,000 prompts. As illustrated in Figure 4, we observe a statistically significant negative correlation (Spearman $\rho = -0.4$, $p < 0.001$): prompts initialized in high-potential regions consistently yield lower diversity, whereas low-potential regions preserve semantic variation. This phenomenon is further verified by our toy experiment in Figure 3, where standard Gaussian initialization collapses into the dominant central mode, leaving non-dominant modes undiscovered.

These findings motivate our proposed approach that explores diversity by actively reshaping the initialization distribution. By biasing the latent prior toward the low-potential regions of the landscape, we can effectively disperse sampling trajectories and ensure wide mode coverage, as shown in Figure 3.

### 4.2. Problem Formulation

We reframe choosing the initial noise not as drawing from a fixed prior, but as sampling from a target *diversity-weighted guidance potential posterior*, $p_{\text{diverse}}(\mathbf{x}_T|c)$. This distribution modulates the standard isotropic Gaussian prior $p(\mathbf{x}_T) = \mathcal{N}(\mathbf{x}_T; \mathbf{0}, \mathbf{I})$ using an energy-based term that promotes diversity:

$$
\begin{aligned}
p_{\text{diverse}}(\mathbf{x}_T|c) &\propto \exp\left(-\tau \cdot U(\mathbf{x}_T, c)\right) \cdot p(\mathbf{x}_T) \\
&\propto \exp\left(-\tau \cdot U(\mathbf{x}_T, c)\right) \cdot \exp(-\frac{1}{2}\|\mathbf{x}_T\|^2),
\end{aligned}
\tag{4}
$$

where $\tau > 0$ is a temperature hyperparameter controlling the trade-off between prior adherence (keeping the latent norm close to the standard Gaussian prior) and diversity maximization (exploring low-potential regions).

We define the potential $U(\mathbf{x}_T, c)$ as a proxy for the *condi-*

*tional guidance intensity* at initialization. To estimate this, we utilize the generalized Tweedie potential, defined as the Euclidean distance between the conditional and unconditional single-step denoising estimates:

$$
U(\mathbf{x}_T, c) = \|\hat{\mathbf{x}}_0(\mathbf{x}_T, c) - \hat{\mathbf{x}}_0(\mathbf{x}_T, \varnothing)\|_2
\tag{5}
$$

Our formulation of $U(\mathbf{x}_T, c)$ in the estimated data space is driven by practical deployment considerations to ensure `DivIn` remains robust and plug-and-play. In standard generation pipelines, the initial continuous timestep $t = T - 1$ varies depending on the chosen noise scheduler and total number of inference steps (e.g., $t = 981$ for a 50-step vs. $t = 958$ for a 30-step schedule). If we utilized the raw score difference $U = \|\Delta s_\theta(x)\|$ as the potential, the time-dependent scaling factor $\lambda_t$ would be absorbed into the temperature, making the optimal hyperparameter $\tau$ highly brittle to changes in inference steps. By projecting to $\hat{x}_0$ via Tweedie's formula, our metric intrinsically handles this dynamic scaling, keeping the optimal temperature $\tau$ remarkably stable across different inference schedules (see Appendix D.5 for detailed ablations). This projection also unifies the potential computation across diffusion and flow matching architectures into a shared Euclidean pixel space.

Minimizing the potential defined in Equation (5) penalizes initial states where the conditional guidance exerts an excessively strong attraction toward a dominant mode and biases the initialization toward flat regions of the landscape, facilitating the discovery of diverse modes.

### 4.3. Algorithm

Since direct sampling from $p_{\text{diverse}}$ is intractable, we employ *Langevin Dynamics* to iteratively transition samples from the initial Gaussian prior toward the target diversity-weighted posterior. The discrete update rule for the latent state $\mathbf{x}_T$ at iteration $k$ is derived from the score of the target distribution, $\nabla_{\mathbf{x}} \log p_{\text{diverse}}(\mathbf{x}|c)$.

Given that the score of the standard normal prior is $\nabla_{\mathbf{x}} \log p(\mathbf{x}) = -\mathbf{x}$, the gradient of the target log-posterior is given by:

$$
\nabla_{\mathbf{x}} \log p_{\text{diverse}} = -\tau \nabla_{\mathbf{x}} U(\mathbf{x}, c) - \mathbf{x}
\tag{6}
$$

Substituting this into the Langevin equation yields our update rule:

$$
\mathbf{x}_T^{(k+1)} = \mathbf{x}_T^{(k)} - \eta\left(\tau \nabla_{\mathbf{x}_T^{(k)}} U(\mathbf{x}_T^{(k)}, c) + \mathbf{x}_T^{(k)}\right) + \sqrt{2\eta}\boldsymbol{\xi}^{(k)}
\tag{7}
$$

where $\eta$ is the step size and $\boldsymbol{\xi}^{(k)} \sim \mathcal{N}(\mathbf{0}, \mathbf{I})$ denotes the injected Gaussian noise.

This mechanism establishes a dynamic equilibrium between three forces. The diversity force $(-\tau\nabla U)$ drives the latent away from high-potential basins and toward flatter regions of the landscape. The prior constraint $(-\mathbf{x})$ anchors

**Algorithm 1** Diversity-inducing Initialization (DivIn)

---

1: **Input:** Condition $c$, steps $K$, step size $\eta$, temperature $\tau$
2: **Initialize:** $\mathbf{x}_T^{(0)} \sim \mathcal{N}(\mathbf{0}, \mathbf{I})$
3: **for** $k = 0$ **to** $K - 1$ **do**
4:     *// Compute conditional guidance potential*
5:     $\hat{\mathbf{x}}_0(\mathbf{x}_T^{(k)}, c) \leftarrow \text{Estimator}(\mathbf{x}_T^{(k)}, c)$
6:     $\hat{\mathbf{x}}_0(\mathbf{x}_T^{(k)}, \varnothing) \leftarrow \text{Estimator}(\mathbf{x}_T^{(k)}, \varnothing)$
7:     $U \leftarrow \|\hat{\mathbf{x}}_0(\mathbf{x}_T^{(k)}, c) - \hat{\mathbf{x}}_0(\mathbf{x}_T^{(k)}, \varnothing)\|$
8:     *// Langevin Update*
9:     $\boldsymbol{\xi}^{(k)} \sim \mathcal{N}(\mathbf{0}, \mathbf{I})$
10:     $\mathbf{g}_{\text{energy}} \leftarrow \tau \nabla_{\mathbf{x}_T^{(k)}} U + \mathbf{x}_T^{(k)}$
11:     $\mathbf{x}_T^{(k+1)} \leftarrow \mathbf{x}_T^{(k)} - \eta \cdot \mathbf{g}_{\text{energy}} + \sqrt{2\eta} \cdot \boldsymbol{\xi}^{(k)}$
12: **end for**
13: $\mathbf{x}_0 \leftarrow \text{Denoise}(\mathbf{x}_T^{(K)}, c)$
14: **Output:** Generated image $\mathbf{x}_0$

---

Figure 5, while SAIL's deterministic updates drive all latents to collapse into the sharp local minimum, DivIn successfully disperses samples across the low-potential basin.

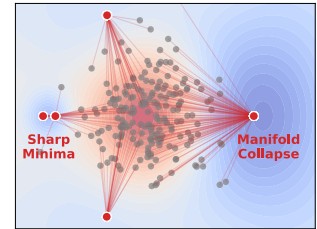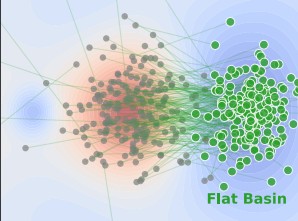

*(a)* Deterministic optimization (SAIL)    *(b)* Stochastic langevin dynamics (Ours)

*Figure 5.* The fundamental distinction between SAIL and DivIn (Ours). The red and green dots represent the optimized initial noise latents $x_T$. While SAIL tends to get trapped in the nearest sharp local minimum (left blue region) and causes volume collapse, DivIn converges to a flat minimum basin (right blue region) while preserving distribution volume.

the latent within the valid manifold of the original normal distribution. The stochastic term ($\sqrt{2\eta}\boldsymbol{\xi}$) injects noise to escape from falling into shallow local minima of the potential landscape, and encourages exploration to optimal regions. While asymptotically exact sampling requires multiple Langevin steps, we empirically find that a single gradient update ($K = 1$) yields significant diversity enhancement. Note that each updating step only introduces an additional forward and backward pass at the initial time step.

The complete procedure is presented in Algorithm 1.

### 4.4. Connection to Sharpness-Aware Initialization

Our framework shares the geometric motivation of Sharpness-Aware Initialization (SAIL) (Jeon et al., 2025), which mitigates memorization based on the sharpness of the log-probability density. However, DivIn introduces two theoretical improvements that address the limitations of SAIL. Firstly, SAIL relies on a Taylor expansion approximation of the Hessian-score product, which requires expensive second-order computations and manually defined thresholds to manage numerical instability. In contrast, we demonstrate in Proposition A.1 that our generalized Tweedie potential $U(\mathbf{x}_T, c)$ serves as a robust proxy for curvature. By formulating this potential in the data space via the estimator $\hat{\mathbf{x}}_0$, our metric naturally handles signal-to-noise scaling, offering superior stability compared to the threshold-dependent metric in SAIL. More fundamentally, SAIL treats seed selection as a *deterministic, mode-seeking optimization* problem, effectively reducing the initialization to a single point estimate. This inevitably collapses the distribution volume and destroys diversity. In contrast, DivIn performs *distributional posterior sampling* over initializations under an explicit Gaussian prior, which is essential for preserving entropy and enabling genuine diversity. As visualized in

## 5. Experiments

In this section, we empirically validate that DivIn achieves a superior diversity-quality Pareto frontier in Section 5.2, and analyze its efficiency, stability, and sensitivity in Sections 5.3 and 5.4. The code is available at https://github.com/South7X/divin.

### 5.1. Experimental Setup

**Datasets and Models** For class-to-image generation, we employ Stable Diffusion (SDv1.4) (Rombach et al., 2022) to produce images for ImageNet-1K (Deng et al., 2009) classes at $256 \times 256$ resolution. We generate a total of 10,000 images using the template prompt "*a photo of a {class name}*" for all classes. The original validation set serves as the reference distribution.

For Text-to-image generation, to evaluate generalizability beyond standard diffusion, we also use the Rectified Flow model Stable Diffusion 3.5 Medium (SDv3.5) (Esser et al., 2024). We follow (Jeon et al., 2025) to use a hybrid general prompt dataset consisting of 500 prompts collected from (1) MS-COCO (Lin et al., 2014) for descriptive captions, (2) Lexica[1] for artistic and stylized prompts, (3) Tuxemon[2] for domain-specific concepts, and (4) GPT-4 (Achiam et al., 2023) generated prompts for complex compositional scenarios. We generate four images per prompt at a resolution of $512 \times 512$.

**Baselines** Besides the base model, we also compare DivIn against a spectrum of training-free diversity en-

---

[1] https://huggingface.co/datasets/vera365/lexica_dataset
[2] https://huggingface.co/datasets/diffusers/tuxemon

*Table 1.* Results of class-to-image generation on ImageNet using SD v1.4. Comparison of our method against initialization-based (SAIL) and trajectory-based baselines (PG, CADS, IG). **Bold** indicates the best result in each group. The results are reported as mean $\pm$ std, computed over 5 independent runs with different seeds on the full dataset.

| Method | Recall ↑ | Vendi Score ↑ | Coverage ↑ | Precision ↑ | Density ↑ | FID ↓ | FD$_{\text{DINOv2}}$ ↓ |
|---|---|---|---|---|---|---|---|
| Base Model | $0.503 \pm 0.009$ | $4.265 \pm 0.008$ | $0.596 \pm 0.004$ | $\mathbf{0.833} \pm 0.006$ | $\mathbf{0.706} \pm 0.007$ | $16.696 \pm 0.062$ | $259.379 \pm 0.987$ |
| + SAIL | $0.543 \pm 0.038$ | $4.549 \pm 0.280$ | $0.591 \pm 0.008$ | $0.825 \pm 0.013$ | $0.677 \pm 0.030$ | $16.395 \pm 0.290$ | $258.376 \pm 3.471$ |
| + DivIn (Ours) | $\mathbf{0.569} \pm 0.013$ | $\mathbf{4.688} \pm 0.121$ | $\mathbf{0.597} \pm 0.006$ | $0.825 \pm 0.010$ | $0.670 \pm 0.011$ | $\mathbf{16.158} \pm 0.299$ | $\mathbf{257.556} \pm 4.311$ |
| PG | $0.502 \pm 0.007$ | $4.395 \pm 0.009$ | $0.604 \pm 0.008$ | $\mathbf{0.835} \pm 0.010$ | $0.704 \pm 0.004$ | $\mathbf{17.520} \pm 0.162$ | $\mathbf{261.782} \pm 1.642$ |
| + DivIn (Ours) | $\mathbf{0.516} \pm 0.008$ | $\mathbf{4.491} \pm 0.029$ | $0.604 \pm 0.005$ | $0.834 \pm 0.008$ | $\mathbf{0.706} \pm 0.009$ | $17.942 \pm 0.341$ | $264.246 \pm 5.713$ |
| CADS | $0.528 \pm 0.008$ | $4.384 \pm 0.014$ | $0.598 \pm 0.008$ | $0.832 \pm 0.002$ | $\mathbf{0.692} \pm 0.011$ | $16.360 \pm 0.160$ | $\mathbf{257.626} \pm 0.774$ |
| + DivIn (Ours) | $\mathbf{0.553} \pm 0.017$ | $\mathbf{4.548} \pm 0.075$ | $\mathbf{0.602} \pm 0.007$ | $0.832 \pm 0.008$ | $0.687 \pm 0.014$ | $\mathbf{16.336} \pm 0.325$ | $258.044 \pm 6.551$ |
| IG | $0.564 \pm 0.006$ | $4.585 \pm 0.016$ | $0.597 \pm 0.002$ | $\mathbf{0.826} \pm 0.013$ | $\mathbf{0.679} \pm 0.006$ | $\mathbf{15.531} \pm 0.054$ | $\mathbf{253.469} \pm 0.476$ |
| + DivIn (Ours) | $\mathbf{0.576} \pm 0.020$ | $\mathbf{4.729} \pm 0.080$ | $\mathbf{0.599} \pm 0.009$ | $0.825 \pm 0.002$ | $0.676 \pm 0.012$ | $15.877 \pm 0.312$ | $256.201 \pm 6.970$ |

*Table 2.* Results of text-to-image generation on general prompts using SD v3.5. The results are reported as mean $\pm$ std, computed over 5 independent runs with different seeds on the full dataset.

| Method | Similarity ↓ | Vendi Score ↑ | CLIP Score ↑ | Aesthetic Score ↑ | ImageReward ↑ |
|---|---|---|---|---|---|
| Base Model | $0.793 \pm 0.002$ | $1.803 \pm 0.008$ | $\mathbf{34.04} \pm 0.04$ | $5.726 \pm 0.006$ | $\mathbf{0.544} \pm 0.010$ |
| + SAIL | $0.780 \pm 0.003$ | $1.850 \pm 0.011$ | $33.97 \pm 0.03$ | $5.734 \pm 0.009$ | $0.536 \pm 0.012$ |
| + DivIn (Ours) | $\mathbf{0.775} \pm 0.002$ | $\mathbf{1.864} \pm 0.007$ | $34.01 \pm 0.02$ | $\mathbf{5.738} \pm 0.014$ | $0.534 \pm 0.010$ |
| CADS | $0.770 \pm 0.003$ | $1.884 \pm 0.014$ | $\mathbf{33.97} \pm 0.05$ | $5.749 \pm 0.006$ | $\mathbf{0.537} \pm 0.014$ |
| + DivIn (Ours) | $\mathbf{0.761} \pm 0.004$ | $\mathbf{1.918} \pm 0.013$ | $33.94 \pm 0.04$ | $\mathbf{5.753} \pm 0.013$ | $0.516 \pm 0.021$ |
| IG | $0.748 \pm 0.002$ | $1.964 \pm 0.008$ | $33.92 \pm 0.03$ | $5.781 \pm 0.004$ | $\mathbf{0.521} \pm 0.009$ |
| + DivIn (Ours) | $\mathbf{0.746} \pm 0.008$ | $\mathbf{1.973} \pm 0.027$ | $\mathbf{33.95} \pm 0.06$ | $\mathbf{5.787} \pm 0.012$ | $0.501 \pm 0.028$ |

hancement methods, categorized by their intervention mechanism. We include trajectory guidance methods, which take effect during the denoising process via particle repulsion or diversity conditioning: Particle Guidance (PG) (Corso et al., 2024), Condition-Annealed Diffusion Sampling (CADS) (Sadat et al., 2024), and Interval Guidance (IG) (Kynkäänniemi et al., 2024). We also compare our method with the initialization optimization method Sharpness-Aware Initialization (SAIL) (Jeon et al., 2025). We treat SAIL as the primary deterministic counterpart to our stochastic approach. For a fair comparison, all reported results use the optimal hyperparameter settings for baselines. Detailed experimental settings and additional results are reported in the Appendix C and D.

**Evaluation Metrics** We consider multi-faceted evaluation metrics to capture generative performance. All results are tracked per class/prompt and averaged. For diversity metrics, we report the Vendi Score (Friedman & Dieng, 2023), which quantifies the effective number of modes. For the class-to-image task, we report recall (Kynkäänniemi et al., 2019) and coverage (Naeem et al., 2020) to assess support overlap with the reference distribution. Following (Corso et al., 2024), we include the in-batch similarity score for text-to-image tasks. To evaluate image quality and fidelity, we use precision (Kynkäänniemi et al., 2019), density (Naeem et al., 2020), FID (Fréchet Inception Distance) (Heusel et al., 2017), FD$_{\text{DINOv2}}$ (Fréchet Distance with DINOv2 features Stein et al. (2023); Oquab et al. (2024)), and Im-

ageReward Xu et al. (2023). We record the CLIP score to quantify the semantic alignment between the generated images and the text prompts.

## 5.2. Main Results

**Quantitative Results** Tables 1 and 2 summarize the performance of DivIn across class-to-image and text-to-image tasks. As shown in Table 1, our initialization-only strategy (*Base + DivIn*) achieves a Vendi score of 4.688, outperforming not only the baseline initialization method (SAIL) but also trajectory-based mechanisms (PG, CADS, and IG). Furthermore, unlike methods that often trade off quality for diversity, DivIn improves image fidelity, achieving the lowest FID (16.158) among all standalone baselines except IG. We observe consistent trends on the SD v3.5 model (Table 2), where DivIn effectively enhances diversity, reducing in-batch similarity and increasing the Vendi score. Notably, this diversity gain does not come at the cost of semantic alignment, as our method maintains a high CLIP score and achieves a higher Aesthetic score than the base model and SAIL, suggesting that the generated samples remain satisfying quality and text alignment.

A key advantage of DivIn is its orthogonality to sampling-time guidance. As shown in the bottom cells of both tables, combining DivIn with trajectory-based methods (PG, CADS, IG) yields a further significant growth in diversity metrics (as indicated by Vendi score, recall, and coverage) with minimal impact on fidelity. This verifies our hypothesis that choosing initial noise from the guidance potential posterior captures diverse modes that are otherwise neglected by standard sampling, even when sophisticated trajectory guidance is applied. By initializing latents in flatter regions of the potential landscape, DivIn naturally complements existing methods to explore rare modes.

**Qualitative Examples** Figure 2 and Figure 7 provide visualized examples demonstrating that DivIn alleviates mode collapse where standard sampling fails. Figure 2 il-

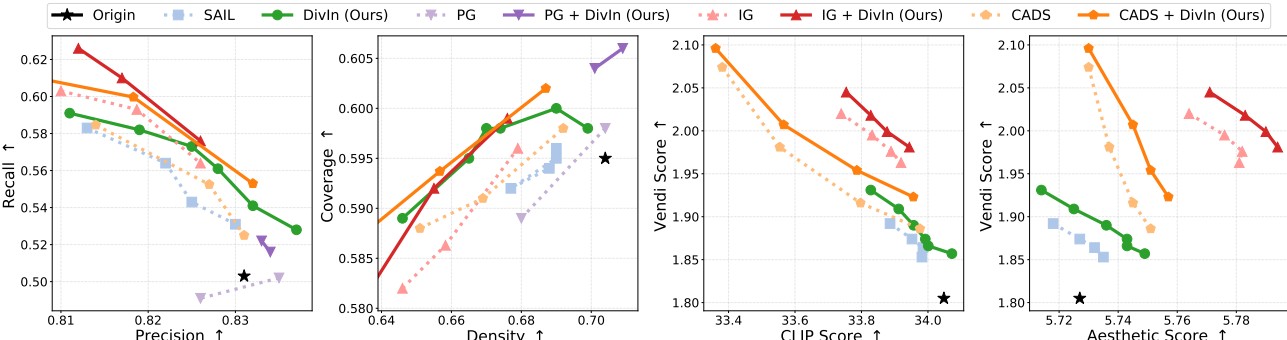

*Figure 6.* Diversity-fidelity Pareto frontiers. The plots illustrate how the hyperparameters of different methods affect the trade-off between image quality (x-axis) and diversity (y-axis) on ImageNet (left) and general prompts (right). Baselines with `DivIn` integrated (solid lines) consistently expand the Pareto frontiers of the baselines alone (dashed lines). Full results in Appendix D.

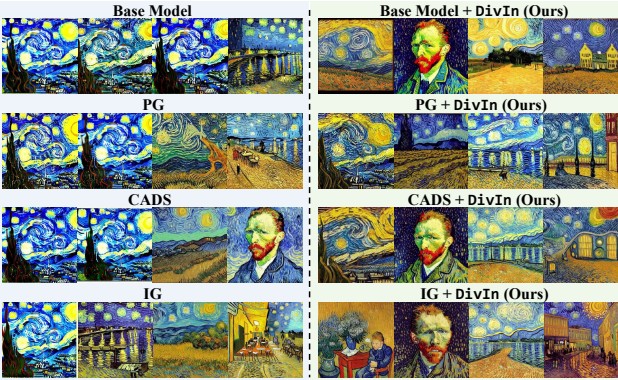

*Figure 7.* Visualizing the orthogonality of `DivIn` to baseline models. We compare standard sampling (Base Model), trajectory-based methods (PG, CADS, IG), and their combinations with `DivIn` on the prompt "*Van Gogh painting*". While baselines tend to collapse to a single dominant copy (first column), plugging in `DivIn` (right four columns) restores diversity across all methods, introducing distinct variations.

lustrates the versatility of `DivIn` across different models and prompts. In the left two prompts (diffusion model), the baseline produces near-duplicate images with identical poses and artwork copies. Our method successfully diversifies the generated images. This improvement also extends to flow matching models. For example, for the "*crystal cave*" prompt, the baseline collapses to a single blue hue, whereas `DivIn` uncovers multi-colored variants (purple, orange, green) while maintaining the prompt's semantics. In Figure 7, we visualize samples for the prompt "*Van Gogh painting*" across different diversity enhancement strategies. Standard sampling and trajectory-intervention baselines (PG, CADS, IG) often converge to a rigid, repetitive generation, dominated by the well-known copy (the first column). By contrast, plugging `DivIn` into these mechanisms consistently discovers diverse modes, generating variations in different art pieces without losing the artistic style. Note that `DivIn` does not eliminate but under-samples the

dominant mode (e.g., "*Starry Night*"). This visually validates that our initialization strategy can further benefit the diversity enhancement on top of various sampling strategies. More generated examples can be found in Appendix E.

**Pareto Frontier Expansion** We further analyze the diversity-quality trade-off by sweeping the hyperparameter settings for all trajectory-based methods and temperature choices for `DivIn` in Figure 6. We observe a consistent pattern of Pareto dominance that the curves depicting methods integrated with `DivIn` (solid lines) consistently encompass their baseline counterparts (dashed lines). Specifically, for any fixed level of image fidelity (measured by precision, density, CLIP score, or aesthetic score), the integration of `DivIn` yields significantly higher diversity (recall, coverage, or Vendi score). This visualizes our finding that initialization with guidance potential posterior unlocks a distinct source of diversity that is additive to interventions during the denoising process. `DivIn` pushes the entire Pareto frontier outward, allowing for more diverse generation without penalty in semantic alignment or image quality.

### 5.3. Analysis on Stochastic Formulation

We analyze the implications of shifting from deterministic SAIL to stochastic Langevin sampling (`DivIn`).

**Manifold Preservation** In Figure 8b, we track the sharpness and mean latent norm $\|\mathbf{x}_T\|_2$ over SAIL optimization process. Note that for an isotropic Gaussian distribution in $d$ dimensions, the expected $L_2$ norm is tightly concentrated around $\sqrt{d}$, which is approximately 128 for our latent space. However, we observe that SAIL suffers from a substantial degradation in Gaussianity (shrinking below 128) because of its deterministic nature, which drives the latent variables off the natural manifold. These out-of-distribution latents lead to severe high-frequency artifacts, as visualized in Figure 9 (Right). We further validate that applying a projection

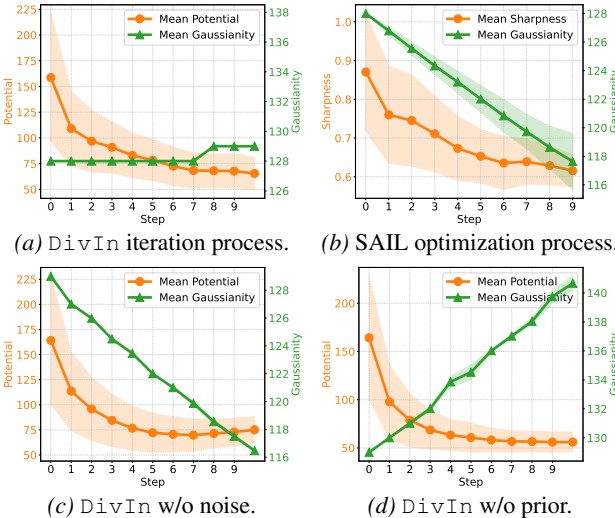

*(a)* `DivIn` iteration process.  *(b)* SAIL optimization process.

*(c)* `DivIn` w/o noise.  *(d)* `DivIn` w/o prior.

*Figure 8.* We track the guidance potential/sharpness (orange, lower is better) and latent Gaussianity (green, mean latent norm $\|x_T\|_2$) over 10 iteration steps. *(a)* `DivIn` successfully minimizes potential while maintaining Gaussianity. *(b)* SAIL optimizes sharpness but drifts the latent norm off-manifold, leading to artifacts. *(c)* Similar to SAIL, without noise injection, the latent collapses. *(d)* Without the prior term, the latent norm explodes. Both an exploding norm and a collapsing norm indicate a shift away from the ideal initial standard Gaussian distribution.

step in SAIL updates can fix the shrinking norm but sacrifices diversity in Appendix D.4. In contrast, the Langevin dynamics of `DivIn` (Figure 8a) balance the diversity gradient with a restorative prior and stochastic noise. This preserves the distributional properties of the initialization to keep the norm consistent with the standard initialization, allowing for deep exploration of the potential landscape without sacrificing image validity.

**Stability and Efficiency**  Figure 9 (left) highlights the robustness of our approach compared to the fragility of SAIL. We find that the performance of SAIL is highly sensitive to its early stopping condition, as a strict threshold can trigger image quality collapses (FID spikes to 27.11), leading to artifacts (as shown in Figure 9 right-top examples) because of the loss of Gaussianity as discussed above. Conversely, `DivIn` exhibits a smooth curve in p and remains stable even after multiple Langevin steps.

Furthermore, as `DivIn` is a training-free method, it incurs zero training overhead, and the additional inference overhead is minimal and highly controllable. As shown in Figure 9 (left), `DivIn` is computationally superior as it requires only $\sim 3\%$ additional wall-clock time per image for a single step ($K = 1$), increasing the generation time from 0.754 seconds to 0.779 seconds. In contrast, SAIL's rejection sampling loop and second-order approximation introduce higher computational cost.

| Model | | Recall ↑ | FID ↓ | Time ↓ | NFE ↓ |
|---|---|---|---|---|---|
| Base model | | 0.503 | 16.70 | 0.754 | 50.00 |
| + SAIL | thres.=7.8 | 0.543 | 16.39 | 0.806 | 52.41 |
| | thres.=7.7 | 0.564 | 16.22 | 0.825 | 52.83 |
| | thres.=7.6 | 0.583 | 16.09 | 0.850 | 53.36 |
| | thres.=7.1 | 0.626 | 27.11 | 1.193 | 60.88 |
| + DivIn | step=1 | 0.569 | 16.16 | 0.779 | 51.00 |
| | step=2 | 0.589 | 16.02 | 0.812 | 52.00 |
| | step=3 | 0.596 | 15.98 | 0.849 | 53.00 |
| | step=10 | 0.609 | 17.02 | 1.089 | 60.00 |

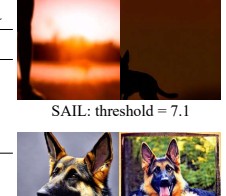

SAIL: threshold = 7.1

Ours: step = 10

*Figure 9.* Left: Quantitative comparison of efficiency and stability. SAIL is unstable as decreasing the threshold causes a catastrophic increase in FID. `DivIn` consistently shows promising results with less additional computational cost. Right: Generations on prompt "*a photo of a German shepherd.*" SAIL results in high-frequency noise artifacts, while `DivIn` still produces natural and high-quality images as optimization steps increase.

### 5.4. Ablation Study

**Effect of Noise and Prior**  Figure 8 and Table 3 study the effect of the stochastic term $\sqrt{2\eta}\xi$ and prior term $\nabla_{\mathbf{x}} \log p(\mathbf{x})$ in the updating rule of `DivIn` defined in Equation (7). We observe a sharp drop in diversity in the w/o noise variant, with recall falling from 0.569 to 0.541 and Vendi score from 4.69 to 4.53. Notably, removing the stochastic term degrades performance to the level of the deterministic baseline (SAIL), empirically proving that the diversity gain is not from the geometric potential alone, but also from the Langevin sampling formulation. Furthermore, without the prior term, diversity metrics degrade relative to the full method, implying that the latent fails to cover the high-density regions of the target distribution without the restorative force. In Table 3, we note a counterintuitive phenomenon that the prior-free setting leads to a better image quality (indicated by FID) than the noise-free setting. We hypothesize that this is because diffusion models react differently to minor off-manifold initial latents when the Langevin step $K = 1$. As we increase steps $K = 3$, image quality degrades significantly for both the noise-free (FID 17.53) and prior-free (FID 17.36) settings compared to the full `DivIn` (FID 15.98). Overall, the full `DivIn` achieves the best balance between the prior and noise to guide the stochastic exploration toward valid and diverse modes.

*Table 3.* Analysis on the updating rule component of `DivIn`. Removing noise (w/o noise) and prior (w/o prior) degrades diversity performance, proving that both prior guidance and noise injection in stochastic sampling are essential in `DivIn` for mode discovery.

| Method | Recall ↑ | Vendi Score ↑ | Precision ↑ | FID ↓ |
|---|---|---|---|---|
| Base Model | 0.503 | 4.265 | **0.833** | 16.696 |
| SAIL (Baseline) | 0.543 | 4.549 | 0.825 | 16.395 |
| `DivIn` w/o noise | 0.541 | 4.534 | 0.822 | 16.544 |
| `DivIn` w/o prior | 0.557 | 4.584 | 0.824 | **16.121** |
| `DivIn` | **0.569** | **4.688** | 0.825 | 16.158 |

*Table 4.* `DivIn`'s performance sensitivity to the Langevin hyper-parameters step size $\eta$ and noise scale $\sqrt{2\eta}$.

| Step size ($\eta$) | Noise scale ($\sqrt{2\eta}$) | Recall ↑ | Precision ↓ |
|---|---|---|---|
| 0.01 | 0.141 | 0.514 | 0.838 |
| 0.05 | 0.316 | 0.569 | 0.825 |
| 0.1 | 0.447 | 0.605 | 0.802 |

| $\tau$ | Recall ↑ | Precision ↑ |
|---|---|---|
| 0.0 | 0.500 | 0.839 |
| 0.6 | 0.569 | 0.825 |
| 1.0 | 0.607 | 0.807 |

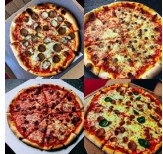 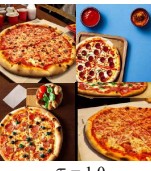

$\tau = 0.0$      $\tau = 1.0$

*Figure 10.* `DivIn`'s performance sensitivity to temperature $\tau$.

**Hyperparameter Sensitivity** We investigate the hyperparameter sensitivity of `DivIn` by scaling the number of steps $K$, step size $\eta$, and temperature $\tau$. For the number of steps ($K$), we provided an ablation in Figure 9 (left table) and Table 8 demonstrating that `DivIn` achieves better diversity with increasing Langevin steps at the expense of degraded FID. We also observe that this trend becomes stable from step 10, when the Langevin dynamics process converges to the stationary distribution of the target posterior. We also conduct an ablation study on the effect of step size ($\eta$) and the noise scale ($\sqrt{2\eta}$), as presented in Table 4, increasing the step size achieves higher Recall by exploring further into the low-potential landscape, with a trade-off in Precision.

In Figure 6 and details in Appendix D, we show the diversity-quality trade-off curves of `DivIn` with different temperatures. We also illustrate the effect of temperature in Figure 10. It clearly shows that increasing $\tau$ from 0.0 to 1.0 yields a substantial gain in recall (from 0.500 to 0.607) with a slight decrease in precision. As visualized, a zero temperature behaves similarly to the base model, where generated samples appear closely similar to each other. In contrast, higher temperatures ($\tau = 1.0$) successfully discover diverse semantic attributes while maintaining structural coherence, demonstrating that `DivIn` offers a wide effective adjusting range compared to SAIL.

## 6. Limitations and Future Work

Despite its efficacy in expanding the diversity-quality Pareto frontier, `DivIn` presents a few limitations. First, our framework inherently relies on the temperature hyperparameter $\tau$ to explicitly control the trade-off between standard prior adherence and diversity-driven exploration. Consequently, deploying `DivIn` on entirely novel generative architectures or highly specialized downstream domains may require empirical tuning to determine the optimal temperature. Second, approximating the target posterior via Langevin dynamics inevitably introduces additional forward and backward

passes at the initial sampling step. While a single updating step ($K = 1$) incurs marginal overhead in practice, achieving asymptotically exact sampling requires multiple iterations, which scales the initialization time. Furthermore, the stochastic nature of Langevin dynamics inherently introduces higher metric variance across independent runs compared to a fixed Gaussian initialization, though its mean performance remains consistently superior. Third, the current formulation of `DivIn` is exclusively applicable to conditional generative models. Because our diversity objective relies on the conditional guidance potential, it cannot natively support unconditional generation, where trajectory-based interventions can apply.

Future work could explore distilling the proposed guidance potential posterior directly into a lightweight, learned prior network to eliminate the need for iterative stochastic Langevin dynamics at inference entirely. Additionally, extending this geometric initialization perspective to unconditional settings, such as by leveraging unconditional score landscape curvature, represents an important direction to fully generalize our framework. Finally, expanding this method to other modalities, such as video generation, where mode collapse remains a critical bottleneck, presents a compelling frontier for future research.

## 7. Conclusion

We introduce *Diversity-Inducing Initialization* (`DivIn`), a plug-and-play algorithm leveraging Langevin dynamics to sample from the *guidance potential posterior* that biases initialization towards low-potential regions where diverse trajectories naturally diverge. Our results verify that `DivIn` significantly expands the diversity-quality Pareto frontier across both diffusion and flow matching architectures and serves as a powerful orthogonal enhancement to existing trajectory-based methods.

## Acknowledgements

The authors are grateful to Srinivas Anumasa for early discussions on the paper. This material is based upon work supported by the Air Force Office of Scientific Research under award number FA2386-24-1-4011, and this research is partially supported by the Singapore Ministry of Education Academic Research Fund Tier 1 (Award No: T1 251RES2509).

## Impact Statement

This paper presents work whose goal is to advance the field of Machine Learning. There are many potential societal consequences of our work, none which we feel must be specifically highlighted here.

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

# A. Theoretical Analysis

In this section, we provide the formal derivation and theoretical justification for our DivIn (Diversity-inducing Initialization). We start with the geometric interpretation in Appendix A.1 that our proposed potential $U(\mathbf{x}_T, c)$ serves as a proxy for the local sharpness of the conditional distribution. In Appendix A.2, we demonstrate that initializing in regions of high sharpness leads to rapid entropy loss during the reverse diffusion process. Conversely, our method preserves diversity by selecting low-potential initial noise.

## A.1. Theoretical Justification: Tweedie Potential as a Sharpness Proxy

We provide a theoretical foundation for using the Generalized Tweedie Potential, defined as $U(\mathbf{x}_t, c) = \|\hat{\mathbf{x}}_0(\mathbf{x}_t, c) - \hat{\mathbf{x}}_0(\mathbf{x}_t, \varnothing)\|$, as a proxy for minimizing landscape sharpness. We establish that minimizing this potential is geometrically equivalent to minimizing the spectral difference between the conditional and unconditional Hessians.

**Proposition A.1** (Tweedie-Hessian Connection). *Minimizing the squared Tweedie difference $\|\hat{\mathbf{x}}_0(\mathbf{x}_t, c) - \hat{\mathbf{x}}_0(\mathbf{x}_t, \varnothing)\|^2$ provides an estimate for minimizing the conditional sharpness. This relationship becomes analytically equivalent to minimizing the weighted trace of the squared difference between the unconditional and conditional Hessians under local Gaussian assumptions.*

*Proof.* The derivation proceeds in two stages: establishing the relationship between the Tweedie update and the score difference, and subsequently linking the score difference to the Hessian trace.

Firstly, we relate the Tweedie potential to the score difference. Recall the Tweedie formula for estimating the clean data $\mathbf{x}_0$ from a noisy latent $\mathbf{x}_t$ in a standard diffusion process with noise schedule $\bar{\alpha}_t$:

$$\hat{\mathbf{x}}_0(\mathbf{x}_t) = \frac{\mathbf{x}_t - \sqrt{1 - \bar{\alpha}_t}\boldsymbol{\epsilon}_\theta(\mathbf{x}_t)}{\sqrt{\bar{\alpha}_t}}. \tag{8}$$

The potential $U(\mathbf{x}_t, c)$ measures the geometric discrepancy between the conditional and unconditional estimators. Substituting the score function relation $\boldsymbol{\epsilon}_\theta(\mathbf{x}_t) \approx -\sqrt{1 - \bar{\alpha}_t}\nabla_{\mathbf{x}_t} \log p_t(\mathbf{x}_t)$, we obtain:

$$\begin{aligned}
\Delta\hat{\mathbf{x}}_0 &= \hat{\mathbf{x}}_0(\mathbf{x}_t, c) - \hat{\mathbf{x}}_0(\mathbf{x}_t, \varnothing) \\
&= -\frac{\sqrt{1 - \bar{\alpha}_t}}{\sqrt{\bar{\alpha}_t}}\left(\boldsymbol{\epsilon}_\theta(\mathbf{x}_t, c) - \boldsymbol{\epsilon}_\theta(\mathbf{x}_t, \varnothing)\right) \\
&= \frac{1 - \bar{\alpha}_t}{\sqrt{\bar{\alpha}_t}}\left(\nabla_{\mathbf{x}_t} \log p_t(\mathbf{x}_t|c) - \nabla_{\mathbf{x}_t} \log p_t(\mathbf{x}_t)\right),
\end{aligned} \tag{9}$$

where score difference $\mathbf{s}^\Delta(\mathbf{x}_t) = \nabla_{\mathbf{x}_t} \log p_t(\mathbf{x}_t|c) - \nabla_{\mathbf{x}_t} \log p_t(\mathbf{x}_t)$. Letting $\lambda_t = \frac{1 - \bar{\alpha}_t}{\sqrt{\bar{\alpha}_t}}$ denote the signal-to-noise ratio scaling factor, the squared potential is strictly proportional to the norm of the score difference:

$$\|\Delta\hat{\mathbf{x}}_0\|^2 = \lambda_t^2 \|\mathbf{s}(\mathbf{x}_t, c) - \mathbf{s}(\mathbf{x}_t)\|^2. \tag{10}$$

Next, we connect the score difference to Hessian sharpness. We assume that the unconditional and conditional distributions at timestep $t$ can be locally approximated by Gaussians $p_t(\mathbf{x}) \approx \mathcal{N}(\boldsymbol{\mu}, \boldsymbol{\Sigma})$ and $p_t(\mathbf{x}|c) \approx \mathcal{N}(\boldsymbol{\mu}_c, \boldsymbol{\Sigma}_c)$. Using Lemma 4.2 from Jeon et al. (2025), which decomposes the expected squared score difference into a first-order mean shift and a second-order curvature mismatch:

$$\mathbb{E}_{p(\mathbf{x}|c)}\left[\|\mathbf{s}^\Delta(\mathbf{x})\|^2\right] = \|H(\mathbf{x})(\boldsymbol{\mu} - \boldsymbol{\mu}_c)\|^2 + \text{tr}\left((H(\mathbf{x}) - H_c(\mathbf{x}))^2 H_c^{-1}(\mathbf{x})\right), \tag{11}$$

where $H(\mathbf{x}) = \boldsymbol{\Sigma}^{-1}$ and $H_c(\mathbf{x}) = \boldsymbol{\Sigma}_c^{-1}$ represent the Hessians (sharpness matrices) of the log-densities.

Additionally, considering the case where the local approximations share similar means ($\boldsymbol{\mu} \approx \boldsymbol{\mu}_c$) and have commuting covariances, the relation simplifies to the spectral difference:

$$\mathbb{E}\left[\|\mathbf{s}^\Delta(\mathbf{x})\|^2\right] \approx \sum_{i=1}^d \frac{(\nu_i - \nu_{i,c})^2}{\nu_{i,c}}, \tag{12}$$

where $\nu_i$ and $\nu_{i,c}$ are the eigenvalues of the unconditional and conditional Hessians, respectively.

Combining Eq. (10) and Eq. (12), we obtain:

$$\mathbb{E}[\|\Delta\hat{\mathbf{x}}_0\|^2] \propto \lambda_t^2 \operatorname{tr}\left((H - H_c)^2 H_c^{-1}\right). \tag{13}$$

This derivation establishes that minimizing the Tweedie Potential $U(\mathbf{x}_t, c)$ serves as a proxy for minimizing the spectral gap between the conditional and unconditional landscapes. Geometrically, this penalizes initialization in regions where the conditioning $c$ induces extreme sharpness ($\nu_{i,c} \gg \nu_i$) relative to the prior.

$\square$

### A.1.1. EXTENSION TO FLOW MATCHING MODELS

We demonstrate that the geometric principles derived above extend to Flow Matching (FM) models (e.g., Stable Diffusion 3), where the generative process is governed by an Ordinary Differential Equation (ODE).

**Proposition A.2** (Flow-Hessian connection). *For Flow Matching models with linear interpolation paths, minimizing the velocity projection error $\|\Delta\hat{\mathbf{x}}_0\|_{FM}$ serves as a proxy for the score difference norm. Consequently, minimizing this error is equivalent to minimizing the conditional sharpness.*

*Proof.* In Flow Matching, the forward process is defined by a linear interpolation $\mathbf{x}_t = (1-t)\mathbf{x}_0 + t\mathbf{x}_1$, where $\mathbf{x}_1 \sim \mathcal{N}(\mathbf{0}, \mathbf{I})$. The model learns a velocity field $\mathbf{v}_\theta(\mathbf{x}_t, t)$ approximating the time derivative $\dot{\mathbf{x}}_t = \mathbf{x}_1 - \mathbf{x}_0$. The clean data $\hat{\mathbf{x}}_0$ can be recovered geometrically by projection:

$$\hat{\mathbf{x}}_0^{\text{flow}}(\mathbf{x}_t) = \mathbf{x}_t - t \cdot \mathbf{v}_\theta(\mathbf{x}_t). \tag{14}$$

We define the FM-Tweedie Potential as the magnitude of the velocity shift induced by conditioning:

$$\begin{aligned}
\Delta\hat{\mathbf{x}}_0 &= \hat{\mathbf{x}}_0(\mathbf{x}_t, c) - \hat{\mathbf{x}}_0(\mathbf{x}_t, \varnothing) \\
&= (\mathbf{x}_t - t \cdot \mathbf{v}_\theta(\mathbf{x}_t, c)) - (\mathbf{x}_t - t \cdot \mathbf{v}_\theta(\mathbf{x}_t, \varnothing)) \\
&= -t\left(\mathbf{v}_\theta(\mathbf{x}_t, c) - \mathbf{v}_\theta(\mathbf{x}_t, \varnothing)\right).
\end{aligned} \tag{15}$$

To relate this velocity difference to the Hessian, we utilize the correspondence between the flow-based estimator and the score-based estimator. For Gaussian probability paths, Tweedie's formula provides a score-based estimate of the clean data:

$$\hat{\mathbf{x}}_0^{\text{score}}(\mathbf{x}_t) \approx \mathbf{x}_t + \sigma^2(t)\nabla_{\mathbf{x}} \log p_t(\mathbf{x}_t). \tag{16}$$

By equating the geometric flow estimator to the score-based Tweedie estimator ($\hat{\mathbf{x}}_0^{\text{flow}} \approx \hat{\mathbf{x}}_0^{\text{score}}$), we establish a proportionality between the learned velocity field and the score function:

$$-t \cdot \mathbf{v}_\theta(\mathbf{x}_t) \approx \sigma^2(t)\nabla_{\mathbf{x}} \log p_t(\mathbf{x}_t) \implies \mathbf{v}_\theta \propto -\mathbf{s}_\theta. \tag{17}$$

Substituting this relationship into the potential definition yields:

$$\|\Delta\hat{\mathbf{x}}_0\|_{\text{FM}}^2 = t^2\|\mathbf{v}^\Delta(\mathbf{x}_t)\|^2 \propto \lambda_{\text{FM}}^2\|\mathbf{s}(\mathbf{x}_t, c) - \mathbf{s}(\mathbf{x}_t)\|^2. \tag{18}$$

Finally, based on the result from Eq. (12), we conclude that minimizing $\|\Delta\hat{\mathbf{x}}_0\|_{\text{FM}}$ in flow-based models equivalently minimizes the conditional sharpness. This penalty effectively prevents initialization in regions where the condition $c$ imposes excessive curvature, ensuring diverse manifold exploration. $\square$

Furthermore, for fixed timesteps in flow matching models, our potential formulation based on the estimator is exactly proportional to the velocity difference. This exact equality aligns with the theoretical relationships established by (Zheng et al., 2023), confirming that our metric natively translates to velocity-based frameworks.

## A.2. Preservation of Diversity via Entropy Dynamics

We now establish a formal bridge between the geometric notion of *sharpness* (derived in Section A.1) and the statistical notion of *diversity*, quantified here by the differential entropy $\mathcal{H}(p_t)$. We demonstrate that regions of high conditional curvature are the primary drivers of rapid entropy loss—and consequently, mode collapse—during the generative process.

Consider the deterministic Probability Flow ODE formulation of the reverse diffusion process (Song et al., 2021b), whose marginal distributions $\{p_t\}_{t=0}^{T}$ coincide with those of the stochastic SDE:

$$d\mathbf{x} = \mathbf{f}_t(\mathbf{x})dt = \left[-\frac{1}{2}\beta_t\mathbf{x} - \frac{1}{2}\beta_t\nabla_{\mathbf{x}}\log p_t(\mathbf{x}|c)\right]dt, \tag{19}$$

where time $t$ flows backward from $T \to 0$, and $\nabla_{\mathbf{x}}\log p_t(\mathbf{x}|c)$ is the conditional score function.

**Theorem A.3** (Generative entropy evolution). *Let $\{\mathbf{x}_t\}_{t\in[0,T]}$ be the trajectory determined by the Probability Flow ODE of a Variance Preserving (VP) diffusion process. Defined in the generative direction where time flows from $T \to 0$ (i.e., $dt < 0$), the instantaneous rate of change of the conditional differential entropy $\mathcal{H}_t(\mathbf{x}|c)$ with respect to the time step $t$ is:*

$$\frac{d\mathcal{H}_t}{dt} = -\frac{d}{2}\beta_t + \frac{1}{2}\beta_t\mathbb{E}_{p_t(\mathbf{x})}[\mathrm{tr}(H_c(\mathbf{x}))], \tag{20}$$

*where $d$ is the dimensionality of the data space (i.e., $\mathbf{x} \in \mathbb{R}^d$). $H_c(\mathbf{x}) = -\nabla_{\mathbf{x}}^2\log p_t(\mathbf{x}|c)$ is the Hessian of the conditional log-density (local sharpness).*

*Proof.* By the instantaneous change of variables formula (Chen et al., 2018), the time evolution of the log-density under the continuous flow defined by Eq. (19) satisfies:

$$\frac{d\log p_t(\mathbf{x}_t)}{dt} = -\,\mathrm{div}(\mathbf{f}_t(\mathbf{x}_t)). \tag{21}$$

The rate of change of the entropy is the expectation of this divergence:

$$\frac{d\mathcal{H}_t}{dt} = -\frac{d}{dt}\mathbb{E}_{p_t}[\log p_t(\mathbf{x}|c)] = \mathbb{E}_{p_t}[\mathrm{div}(\mathbf{f}_t(\mathbf{x}))]. \tag{22}$$

We compute the divergence of the drift vector field $\mathbf{f}_t$ for the VP-SDE:

$$\begin{aligned}
\mathrm{div}(\mathbf{f}_t) &= \mathrm{div}\left(-\frac{1}{2}\beta_t\mathbf{x} - \frac{1}{2}\beta_t\nabla_{\mathbf{x}}\log p_t(\mathbf{x}|c)\right) \\
&= -\frac{1}{2}\beta_t\,\mathrm{div}(\mathbf{x}) - \frac{1}{2}\beta_t\,\mathrm{div}(\nabla_{\mathbf{x}}\log p_t(\mathbf{x}|c)) \\
&= -\frac{d}{2}\beta_t - \frac{1}{2}\beta_t\Delta\log p_t(\mathbf{x}|c).
\end{aligned} \tag{23}$$

Recognizing that the Laplacian of the log-density corresponds to the negative trace of the Hessian, $\Delta\log p = \mathrm{tr}(\nabla^2\log p) = -\mathrm{tr}(H_c)$, we substitute this back to obtain:

$$\mathrm{div}(\mathbf{f}_t) = -\frac{d}{2}\beta_t + \frac{1}{2}\beta_t\,\mathrm{Tr}(H_c(\mathbf{x})). \tag{24}$$

Taking the expectation over $p_t(\mathbf{x}|c)$, we have the theorem statement. $\square$

**Interpretation**  Theorem A.3 establishes a direct geometric constraint on diversity preservation. Since the generative process flows backward in time ($T \to 0$, implying $dt < 0$), the accumulated change in entropy is determined by the integral of the instantaneous rate $d\mathcal{H}/dt$. Crucially, the equation reveals that the rate of entropy reduction is driven by the local sharpness $\mathrm{tr}(H_c(\mathbf{x}))$. Trajectories initialized in or passing through regions with a large Hessian trace (high sharpness) yield a large positive value for $d\mathcal{H}/dt$. When integrated over the negative time increment $dt$, this results in significant entropy loss, effectively collapsing the distribution volume rapidly. By Proposition A.1, our proposed potential $U(\mathbf{x}_T, c)$ serves as a proxy for the sharpness measure. Sampling from the diversity-weighted initialization $p_{\text{diverse}} \propto e^{-\tau U}$ explicitly biases the initial distribution toward low-potential (smooth) regions. By selecting trajectories that begin in flatter regions of the landscape, we minimize the instantaneous rate of entropy decay, thereby preserving distribution volume and diversity throughout the generative process.

### A.2.1. CONNECTION TO MAXIMUM ENTROPY MANIFOLD EXPLORATION

We now connect our geometric analysis to the Maximum Entropy Manifold Exploration framework (De Santi et al., 2025). This framework seeks to maximize the entropy of the generated data distribution $\mathcal{H}(p_0)$ by fine-tuning model parameters. We demonstrate that our method achieves a similar objective via our initialization strategy DivIn.

**Corollary A.4** (DivIn maximizes a lower bound on data entropy). *From Theorem A.3, the final data entropy is determined by the initial entropy minus the accumulated entropy decay along the trajectory:*

$$\mathcal{H}(p_0) = \mathcal{H}(p_T) - \int_0^T \frac{1}{2}\beta_t \mathbb{E}_{p_t}\left[\text{tr}(H_c(\mathbf{x}_t)) - d\right]dt. \tag{25}$$

*Directly maximizing $\mathcal{H}(p_0)$ is intractable, so we construct a tractable surrogate objective. Our sampling distribution $p_{diverse}(\mathbf{x}_T) \propto \exp(-\frac{1}{2}\|\mathbf{x}_T\|^2 - \tau U(\mathbf{x}_T|c))$ is the unique solution to the following optimization problem:*

$$\max_{p_T}\left\{\mathcal{H}(p_T) - \mathbb{E}_{p_T}\left[\frac{1}{2}\|\mathbf{x}_T\|^2 + \tau U(\mathbf{x}_T|c)\right]\right\}. \tag{26}$$

*By solving this variational problem, DivIn effectively maximizes a lower bound on the final data entropy $\mathcal{H}(p_0)$, preserving diversity by structurally avoiding the high entropy decay identified in Eq.* (25).

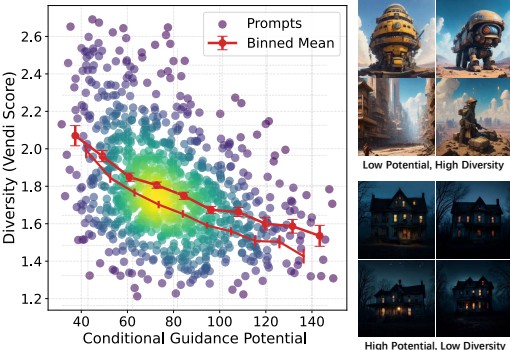

*Figure 11.* In addition to Figure 4, we analyze the initialization potential against generative diversity on SDv3.5 using the general prompts. The binned mean (red line) shows a strong negative correlation (Spearman $\rho = -0.40$) that initialization in high-potential regions leads to rigid mode collapse, while low-potential regions preserve diversity.

## B. Details of Toy Experiment

This section provides additional details on the toy experiment discussed in 4.1. We construct an unbalanced 2D Mixture of Gaussian landscape with a total of 9 modes. A dominant mode is centered at the origin with a larger weight and smaller covariance scales, surrounded by 8 symmetric minor modes with lower, varying weights and slightly larger variance. The unconditional distribution is a uniform mixture over the same means. Note that our design choice for this toy experiment was not intended to be a strict global statistical simulation, but rather an idealized local approximation built for controlled geometric visualization. In the neighborhood of a target conditional mode, the unconditional landscape can be generally much flatter than the sharp basin of attraction formed by the conditional distribution $p(x|c)$. By modeling the unconditional distribution $p(x)$ as uniform, we establish a control variable to approximate this relative and localized flatness. This decoupling allows us to purely isolate the impact of the conditioning signal, visually demonstrating how the sharp geometry of $p(x|c)$ alone induces mode collapse.

We use score-based sampling with Classifier-Free Guidance scale of 7.5 over $T = 50$ steps. We compute exact analytical scores for both conditional and unconditional distributions to isolate sampling dynamics from estimation error. We compare baseline initialization on $N = 100$ initial latents with our DivIn initialization in $K = 2$ steps of Langevin dynamics.

## C. Details of the Main Experiments

This section describes the experimental settings for the main experiments presented in Section 5. All experiments were performed on NVIDIA A100 GPUs. We use Stable Diffusion v1.4 and v3.5 Medium. We use the DDIM sampler (Song

et al., 2021a) with 50 sampling steps and classifier-free guidance scale of 7.5 for SDv1.4, and the default flow-matching Euler scheduler with 30 sampling steps and guidance scale of 7.0 for SDv3.5.

**Details of Baseline Methods**

- **Particle Guidance (Corso et al., 2024)**: This work addressed the limitation that standard sampling generates independent and identically distributed (i.i.d.) samples, which often collapse to single modes. They proposed sampling a set of particles jointly by introducing a repulsive potential to the score function to enforce diversity among the batch of samples. The modified update rule for the $i$-th particle $\mathbf{x}_t^{(i)}$ includes a gradient of a kernel $k$ (e.g., RBF) measuring similarity to other particles:

$$\hat{\epsilon}(\mathbf{x}_t^{(i)}) \leftarrow \hat{\epsilon}(\mathbf{x}_t^{(i)}) - w \sum_{j \neq i} \nabla_{\mathbf{x}_t^{(i)}} k(\mathbf{x}_t^{(i)}, \mathbf{x}_t^{(j)})$$

where $w$ controls the strength of the repulsive force, effectively pushing particles apart in the data manifold.

- **CADS (Sadat et al., 2024)**: This paper identified that the static conditioning signal significantly constrains diversity, particularly at high guidance scales. To mitigate this, they proposed Condition-Annealed Diffusion Sampling (CADS), which dynamically adds Gaussian noise to the conditioning embedding $c$ during the reverse process. The perturbed condition $c_t'$ is defined as:

$$c_t' = c + \lambda_t \xi, \quad \xi \sim \mathcal{N}(0, \mathbf{I})$$

where $\lambda_t$ is a time-dependent annealing schedule that starts high to encourage diversity and decays to zero to ensure semantic fidelity by the end of generation.

- **Interval Guidance (Kynkäänniemi et al., 2024)**: This paper observed that Classifier-Free Guidance (CFG) is primarily beneficial only during the intermediate noise levels of the diffusion process. They proposed applying guidance only within a specific interval $[t_{\text{start}}, t_{\text{end}}]$. Outside this interval, standard conditional sampling is used to avoid burn-in artifacts (at high noise) and reduce computational cost (at low noise). The guided noise prediction becomes:

$$\tilde{\epsilon}_\theta(x_t, c) = \epsilon_\theta(x_t, \emptyset) + s \cdot \mathbb{K}_{t \in [t_{\text{start}}, t_{\text{end}}]} (\epsilon_\theta(x_t, c) - \epsilon_\theta(x_t, \emptyset))$$

- **SAIL (Jeon et al., 2025)**: This work proposed a geometric framework that links memorization to the sharpness of the probability landscape, quantified by the eigenvalues of the Hessian of the log-density $\nabla_{x_t}^2 \log p_t(x_t)$. They presented that memorized samples correspond to sharp peaks in the distribution. To detect memorization at the earliest stage of generation (e.g., $t = T - 1$), they introduced a curvature-aware metric that amplifies high-curvature directions by weighting the score function with the Hessian. The proposed detection metric is defined as:

$$\|H_\theta^\Delta(x_t, c) s_\theta^\Delta(x_t, c)\|^2,$$

where $s_\theta^\Delta(x_t, c) = s_\theta(x_t, c) - s_\theta(x_t)$ is the difference between conditional and unconditional scores, and $H_\theta^\Delta(x_t, c)$ is the corresponding Hessian difference. This metric effectively captures the extent to which conditioning induces sharpness in the learned distribution. To mitigate memorization, they proposed *Sharpness-Aware Initialization for Latent Diffusion* (SAIL), an inference-time strategy that optimizes the initial noise vector $x_T$ to steer the sampling trajectory toward smoother probability regions:

$$\mathcal{L}_{\text{SAIL}}(x_T) := \|s_\theta^\Delta(x_T + \delta s_\theta^\Delta(x_T)) - s_\theta^\Delta(x_T)\|^2 + \alpha \|x_T\|^2.$$

Here, the first term approximates the Hessian-score product via Taylor expansion to avoid computationally expensive backpropagation through the Hessian, and $\alpha$ balances the regularization. By updating $x_T$ to minimize this loss, SAIL reduces memorization without altering model parameters or the user prompt.

Our `DivIn` is fundamentally distinct from SAIL when generalizing this geometrical motivation to enhancing general diversity. Instead of deterministic optimization, we leverage Langevin dynamics to perform *distributional posterior sampling*, which is essential for maintaining entropy and enabling diversity discovery.

**Details for Our Method** We set Langevin step $K = 1$ for all results except those in Figure 9&8, where we evaluate the sensitivity of steps over a range of [1,10]. We set the step size of 0.05 for Stable Diffusion v1.4 and 0.01 for v3.5. We investigate temperature $\tau \in \{0.2, 0.3, 0.4, 0.6, 0.7, 0.8\}$ for v1.4 and $\{0.09, 0.1, 0.12, 0.14, 0.16, 0.18\}$ for v3.5.

*Table 5.* Results of all methods on ImageNet prompts using the SDv1.4 model in the tradeoff experiments in Figure 6.

| Method | Recall ↑ | Vendi ↑ | Coverage ↑ | Precision ↑ | Density ↑ | FID ↓ | FD$_{DINOv2}$ ↓ |
|---|---|---|---|---|---|---|---|
| Base Model | 0.503 | 4.268 | 0.595 | 0.831 | 0.704 | 16.730 | 259.143 |
| SAIL,threshold=7.6 | 0.583 | 4.687 | 0.596 | 0.813 | 0.690 | 16.090 | 256.236 |
| SAIL,threshold=7.7 | 0.564 | 4.553 | 0.594 | 0.822 | 0.688 | 16.223 | 255.626 |
| SAIL,threshold=7.8 | 0.543 | 4.549 | 0.592 | 0.825 | 0.677 | 16.368 | 256.565 |
| SAIL,threshold=7.9 | 0.531 | 4.396 | 0.595 | 0.830 | 0.690 | 16.434 | 256.695 |
| DivIn (Ours), temperature=0.2 | 0.528 | 4.416 | 0.598 | 0.837 | 0.699 | 16.375 | 256.036 |
| DivIn (Ours), temperature=0.3 | 0.541 | 4.496 | 0.600 | 0.832 | 0.690 | 16.218 | 255.123 |
| DivIn (Ours), temperature=0.4 | 0.561 | 4.563 | 0.598 | 0.828 | 0.674 | 16.176 | 254.599 |
| DivIn (Ours), temperature=0.6 | 0.573 | 4.665 | 0.598 | 0.825 | 0.670 | 16.026 | 255.008 |
| DivIn (Ours), temperature=0.7 | 0.582 | 4.711 | 0.595 | 0.819 | 0.665 | 16.029 | 255.471 |
| DivIn (Ours), temperature=0.8 | 0.591 | 4.761 | 0.589 | 0.811 | 0.646 | 15.951 | 256.385 |
| Particle Guidance,strength=32 | 0.502 | 4.395 | 0.598 | 0.835 | 0.704 | 16.636 | 256.670 |
| Particle Guidance,strength=64 | 0.491 | 4.621 | 0.589 | 0.826 | 0.680 | 23.185 | 286.467 |
| PG+DivIn (Ours), temperature=0.1 | 0.522 | 4.407 | 0.606 | 0.833 | 0.709 | 16.542 | 253.186 |
| PG+DivIn (Ours), temperature=0.1 | 0.516 | 4.491 | 0.604 | 0.834 | 0.701 | 17.749 | 259.101 |
| CADS, scale =0.001, $\tau_1$=0.7 | 0.585 | 4.826 | 0.588 | 0.814 | 0.651 | 16.471 | 260.757 |
| CADS, scale =0.001, $\tau_1$=0.8 | 0.553 | 4.584 | 0.591 | 0.827 | 0.669 | 16.205 | 258.149 |
| CADS, scale =0.001, $\tau_1$=0.9 | 0.525 | 4.397 | 0.598 | 0.831 | 0.692 | 16.203 | 257.085 |
| CADS+DivIn (Ours), temperature=0.2, $\tau_1$=0.7 | 0.616 | 5.114 | 0.584 | 0.801 | 0.622 | 16.807 | 260.552 |
| CADS+DivIn (Ours), temperature=0.2, $\tau_1$=0.8 | 0.600 | 4.839 | 0.594 | 0.818 | 0.657 | 16.090 | 254.453 |
| CADS+DivIn (Ours), temperature=0.2, $\tau_1$=0.9 | 0.553 | 4.646 | 0.602 | 0.832 | 0.687 | 16.000 | 252.767 |
| Interval Guidance,[0.1,0.6] | 0.603 | 4.855 | 0.582 | 0.810 | 0.646 | 15.680 | 254.467 |
| Interval Guidance,[0.1,0.7] | 0.593 | 4.743 | 0.586 | 0.819 | 0.658 | 15.633 | 254.694 |
| Interval Guidance,[0.1,0.9] | 0.564 | 4.585 | 0.596 | 0.826 | 0.679 | 15.530 | 253.292 |
| IG+DivIn (Ours), temperature=0.2,[0.1,0.6] | 0.626 | 5.052 | 0.581 | 0.812 | 0.635 | 15.838 | 255.009 |
| IG+DivIn (Ours), temperature=0.2,[0.1,0.7] | 0.610 | 4.915 | 0.592 | 0.817 | 0.655 | 15.749 | 253.060 |
| IG+DivIn (Ours), temperature=0.2,[0.1,0.9] | 0.576 | 4.766 | 0.599 | 0.826 | 0.676 | 15.678 | 251.612 |

*Table 6.* Results of all methods on general prompts using the SDv3.5 model in the tradeoff experiments in Figure 6.

| Method | Similarity ↓ | Vendi Score ↑ | CLIP Score ↑ | Aesthetic Score ↑ | ImageReward ↑ | Time | NFE |
|---|---|---|---|---|---|---|---|
| Base Model | 0.792 | 1.805 | 34.048 | 5.727 | 0.542 | 1.455 | 30.00 |
| SAIL, threshold=325.0 | 0.766 | 1.892 | 33.886 | 5.718 | 0.504 | 2.194 | 36.53 |
| SAIL, threshold=330.0 | 0.772 | 1.874 | 33.952 | 5.727 | 0.519 | 2.046 | 35.47 |
| SAIL, threshold=335.0 | 0.776 | 1.864 | 33.985 | 5.732 | 0.528 | 1.932 | 34.65 |
| SAIL, threshold=340.0 | 0.779 | 1.853 | 33.982 | 5.735 | 0.534 | 1.867 | 34.22 |
| DivIn (Ours), temperature=0.09 | 0.777 | 1.857 | 34.072 | 5.749 | 0.552 | 1.672 | 31.00 |
| DivIn (Ours), temperature=0.1 | 0.775 | 1.866 | 34.001 | 5.743 | 0.525 | 1.620 | 31.00 |
| DivIn (Ours), temperature=0.12 | 0.772 | 1.874 | 33.992 | 5.743 | 0.512 | 1.621 | 31.00 |
| DivIn (Ours), temperature=0.14 | 0.768 | 1.890 | 33.957 | 5.736 | 0.505 | 1.619 | 31.00 |
| DivIn (Ours), temperature=0.16 | 0.762 | 1.909 | 33.912 | 5.725 | 0.492 | 1.622 | 31.00 |
| DivIn (Ours), temperature=0.18 | 0.755 | 1.931 | 33.828 | 5.714 | 0.471 | 1.620 | 31.00 |
| CADS, $\tau_1$=0.6 | 0.716 | 2.074 | 33.381 | 5.730 | 0.382 | 1.458 | 30.00 |
| CADS, $\tau_1$=0.7 | 0.743 | 1.981 | 33.554 | 5.737 | 0.445 | 1.459 | 30.00 |
| CADS, $\tau_1$=0.8 | 0.762 | 1.916 | 33.797 | 5.745 | 0.497 | 1.459 | 30.00 |
| CADS, $\tau_1$=0.9 | 0.770 | 1.886 | 33.976 | 5.751 | 0.533 | 1.461 | 30.00 |
| CADS+DivIn (Ours), temperature=0.06, $\tau_1$=0.6 | 0.711 | 2.096 | 33.361 | 5.730 | 0.350 | 1.629 | 31.00 |
| CADS+DivIn (Ours), temperature=0.06, $\tau_1$=0.7 | 0.736 | 2.007 | 33.566 | 5.745 | 0.424 | 1.630 | 31.00 |
| CADS+DivIn (Ours), temperature=0.06, $\tau_1$=0.8 | 0.750 | 1.954 | 33.786 | 5.751 | 0.476 | 1.631 | 31.00 |
| CADS+DivIn (Ours), temperature=0.06, $\tau_1$=0.9 | 0.759 | 1.923 | 33.956 | 5.757 | 0.517 | 1.627 | 31.00 |
| Interval Guidance, [0.1,0.6] | 0.733 | 2.020 | 33.739 | 5.764 | 0.475 | 1.455 | 30.00 |
| Interval Guidance, [0.1,0.7] | 0.740 | 1.995 | 33.832 | 5.776 | 0.496 | 1.454 | 30.00 |
| Interval Guidance, [0.1,0.8] | 0.745 | 1.976 | 33.889 | 5.782 | 0.506 | 1.454 | 30.00 |
| Interval Guidance, [0.1,0.9] | 0.748 | 1.963 | 33.919 | 5.781 | 0.524 | 1.453 | 30.00 |
| IG+DivIn (Ours), temperature=0.04, [0.1,0.6] | 0.726 | 2.045 | 33.754 | 5.771 | 0.469 | 1.622 | 31.00 |
| IG+DivIn (Ours), temperature=0.04, [0.1,0.7] | 0.733 | 2.018 | 33.828 | 5.783 | 0.488 | 1.620 | 31.00 |
| IG+DivIn (Ours), temperature=0.04, [0.1,0.8] | 0.738 | 1.999 | 33.877 | 5.790 | 0.493 | 1.622 | 31.00 |
| IG+DivIn (Ours), temperature=0.04, [0.1,0.9] | 0.743 | 1.981 | 33.944 | 5.794 | 0.506 | 1.622 | 31.00 |

# D. Additional Experimental Results

## D.1. Further Diversity-Quality Trade-offs

In Figure 6, we plot the Pareto frontiers for both ImageNet (class-to-image) and general prompt (text-to-image) datasets. Detailed numerical results corresponding to these trade-off curves are provided in Table 5 and Table 6. These tables list the specific hyperparameter settings (e.g., temperature $\tau$ for `DivIn`, guidance strength for baselines) used to plot the curves, demonstrating that `DivIn` provides consistent diversity improvement across a wide range of settings without catastrophic quality degradation.

## D.2. Quantitative Results of General Prompts on SDv1.4

In addition to Table 2, we extend our evaluation to Stable Diffusion v1.4 on the general prompts dataset to demonstrate the architectural universality of `DivIn`. As detailed in Table 7, `DivIn` invariably increases the Vendi Score across the base model and all trajectory-based baselines (PG, CADS, and IG). For instance, applying `DivIn` to the base model improves the Vendi score from 2.505 to 2.593. Notably, this improvement in diversity does not come at the expense of semantic alignment, as CLIP scores remain comparable to the baselines. This demonstrates that the benefits of our `DivIn` on the general prompts are not specific to flow matching models (as seen in the main text with SD3.5) but transfer effectively to standard diffusion formulations.

*Table 7.* Results on general prompts using SD v1.4. We compare `DivIn` against initialization-based (SAIL) and trajectory-based (PG, CADS, IG) baselines. `DivIn` consistently improves the Vendi Score when applied on top of the base model or combined with other methods, validating its orthogonality to trajectory interventions.

| Method | Similarity ↓ | Vendi Score ↑ | Clip Score ↑ | Aesthetic Score ↑ |
|---|---|---|---|---|
| Base Model | 0.600 | 2.505 | **32.312** | **5.539** |
| + SAIL | 0.576 | 2.583 | 31.853 | 5.530 |
| + `DivIn` | 0.576 | **2.593** | 31.934 | 5.480 |
| PG | 0.585 | 2.556 | **32.176** | **5.487** |
| + `DivIn` | **0.574** | **2.598** | 32.003 | 5.428 |
| CADS | 0.592 | 2.533 | 32.118 | **5.552** |
| + `DivIn` | **0.587** | **2.549** | **32.119** | 5.502 |
| IG | 0.566 | 2.625 | **31.911** | **5.533** |
| + `DivIn` | **0.560** | **2.647** | 31.708 | 5.470 |

## D.3. Quantitative Results with Increased Langevin Steps

The table in Figure 9 shows that for `DivIn`, both recall and FID increase as the number of Langevin steps increases. As shown in Table 8, we extended the evaluation of `DivIn` on SD v1.4 up to 30 steps. We find that rather than diverging, both FID and Recall quickly reach a stable equilibrium after step 10. The observed trend reflects that the Langevin dynamics process converges to the stationary distribution of the target posterior from step 10 to 30. Thus, the metrics stabilize (FID around 17.0, Recall around 0.61) without further degradation.

*Table 8.* Results on `DivIn` with increased Langevin steps. The image quality (as indicated by FID) does not continuously degrade as the steps increase.

| Step ($k$) | 10 | 20 | 30 |
|---|---|---|---|
| Recall ↑ | 0.609 | 0.617 | 0.614 |
| FID ↓ | 17.02 | 17.04 | 16.99 |

## D.4. Applying Projection Step in Optimization

As we noted in Section 5.3, SAIL suffers from a substantial degradation in Gaussianity, which drives the latent variables off the natural manifold. To explicitly enforce the theoretical norm constraint, we consider applying a spherical projection step

after each gradient update in the optimization process:

$$x_T^{(k)} \leftarrow \sqrt{d} \frac{x_T^{(k)}}{\|x_T^{(k)}\|_2},$$

where $d$ is the dimensionality of the latent space. Geometrically, this operation directly pulls the deviating latent variables back onto the correct $\sqrt{d}$-radius hypersphere, ensuring they remain anchored to the isotropic Gaussian prior.

The empirical results in Table 9 demonstrate that while this projection step successfully corrects SAIL's shrinking norm and slightly improves image precision, it fundamentally sacrifices generative diversity, causing the Vendi score to drop significantly (by 0.093). Because of SAIL's deterministic, mode-seeking nature, explicitly enforcing the norm constraint cannot prevent the underlying collapse of the distribution volume. In contrast, applying the exact same projection step to `DivIn` results in a negligible Vendi score decrease (0.009). This difference highlights that `DivIn`'s stochastic Langevin dynamics intrinsically respect the prior and preserve distribution entropy, enabling robust and diverse mode exploration even under strict manifold constraints.

*Table 9.* Effect of the projection step on generative performance. Adding a projection constraint (+ proj) causes a significant drop in diversity for SAIL, whereas `DivIn` intrinsically preserves distribution entropy and maintains a high Vendi Score.

| Method | Vendi Score ↑ | Precision ↑ |
|---|---|---|
| SAIL | 4.549 | 0.825 |
| SAIL + proj | 4.456 | 0.828 |
| DivIn | 4.699 | 0.825 |
| DivIn + proj | 4.679 | 0.835 |

### D.5. Ablation Study: Tweedie-based vs. Score-based Potential

To validate the practical advantage of our Tweedie-based formulation over a raw score-based formulation $U = \|\Delta s_\theta(x)\|$, we compare their hyperparameter robustness under varying inference steps. While theoretically similar for a fixed timestep up to a scaling factor, the score-based formulation requires wildly retuned temperatures $\tau' = \lambda \cdot \tau$ simply by changing the inference steps (e.g., optimal $\tau' = 130.7$ for 30 steps vs. $\tau' = 171.5$ for 50 steps) to achieve comparable performance. Conversely, our Tweedie formulation intrinsically normalizes this scaling factor, maintaining a completely stable optimal temperature ($\tau = 0.6$) regardless of the chosen inference schedule.

*Table 10.* Comparison of Tweedie-based and score-based formulations across different inference steps.

| Inference steps | Tweedie-based | | | Score-based | | |
|---|---|---|---|---|---|---|
| | $\tau$ | Recall ↑ | FID ↓ | $\tau' = \lambda \cdot \tau$ | Recall ↑ | FID ↓ |
| 30 | 0.6 | 0.585 | 16.28 | 130.7 | 0.585 | 16.29 |
| 50 | 0.6 | 0.569 | 16.16 | 171.5 | 0.565 | 16.22 |

### D.6. Ablation Study: Performance on Different Prompt Length

In Figure 12, we demonstrate the diversity enhancement (indicated by Vendi score) and prompt alignment (indicated by CLIP score) over varying prompt length on the general prompt dataset using `DivIn` with updating step $K = 1$. Intuitively, it is easier to discover diverse images fitting short prompts than highly specified long prompts. Notably, we find that `DivIn` enhances the diversity of prompts across all length categories (from 21 to 500+ characters) while preserving the same CLIP score as the base model. This suggests our method remains effective even under tight semantic constraints.

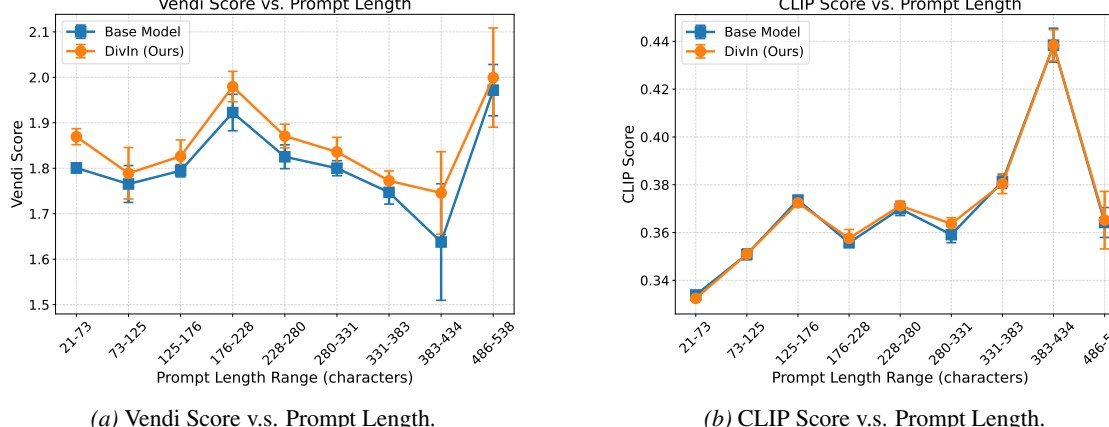

*(a)* Vendi Score v.s. Prompt Length.          *(b)* CLIP Score v.s. Prompt Length.

*Figure 12.* Diversity and prompt alignment for prompts of varying length. We split the general prompts into 10 categories based on the number of characters. We find that `DivIn` achieves a consistently higher diversity than the base base model for both short and long prompts, while keeping the same CLIP score as the base model. An example short prompt is "*A whimsical candy shop*". An example long prompt is "*cinema 4d colorful render, organic, ultra detailed, of a painted realistic glass helmet, scratched. biomechanical cyborg, analog, macro lens, beautiful natural soft rim light, big leaves, winged insects and stems, roots, fine foliage lace, turquoise gold details, Alexander Mcqueen high fashion haute couture, art nouveau fashion embroidered, intricate details, mesh wire, mandelbrot fractal, anatomical, facial muscles, cable wires, elegant, hyper realistic, in front of dark flower pattern wallpaper, ultra detailed, 8k post-production*". Error bars denote the standard error over 5 seeds.

### D.7. Ablation Study: Varying Guidance Weight

The classifier-free guidance scale is a fundamental trade-off parameter between sample fidelity and diversity. In Figure 13, we analyze how `DivIn` interacts with this trade-off on SD v1.4 by sweeping the classifier-free guidance weight $w \in [3.5, 7.5]$. We observe that while increasing guidance weight typically collapses diversity (evidenced by dropping Recall and Vendi Scores), `DivIn` acts as a counterbalance, consistently shifting the diversity curves upwards across the entire range. This indicates that `DivIn` recovers diverse modes in the initialization phase that are otherwise suppressed by strong classifier-free guidance during sampling.

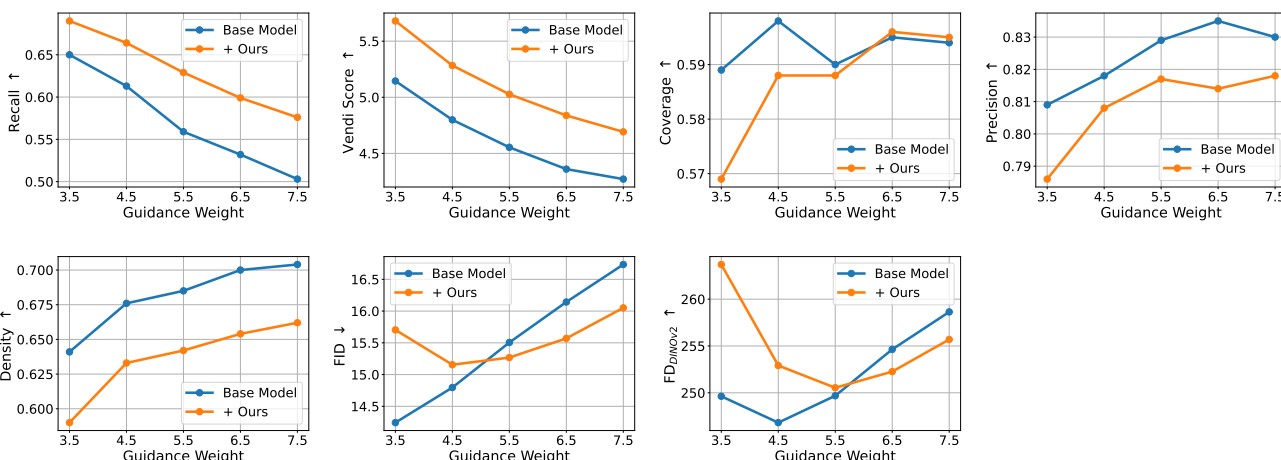

*Figure 13.* Impact of Classifier-Free Guidance (CFG) weight. We evaluate diversity (Recall, Vendi, Coverage) and quality (Precision, Density, FID, $FD_{DINOv2}$) metrics on ImageNet using SD v1.4 while varying the guidance scale. `DivIn` (orange line) consistently outperforms the Base Model (blue line) in diversity metrics across all guidance weights, effectively increasing diversity across varying guidance scales.

# E. Additional Examples of Generated Diverse Images

In addition to the examples in Figure 2, we provide comprehensive qualitative comparisons. Figure 14 to Figure 17 showcase results on SD v3.5 Medium, while Figure 18 to Figure 21 utilize SD v1.4. These visual results demonstrate that DivIn consistently unlocks diverse variations in viewpoint, lighting, composition, and style compared to the highly similar outputs of the base model.

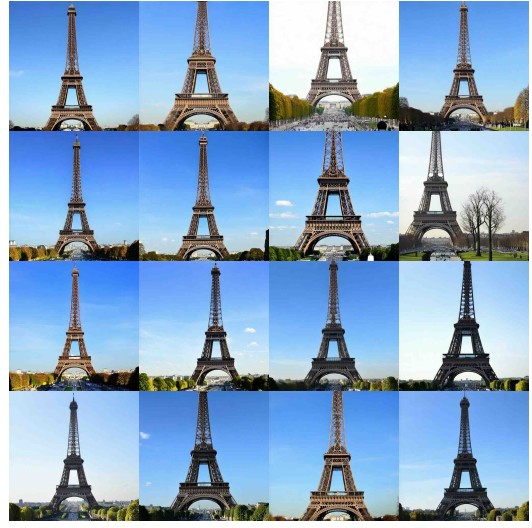

*(a)* Base model without `DivIn`   *(b)* Base model + `DivIn`

*Figure 14.* Images generated with Stable Diffusion v3.5 Medium with and without `DivIn` for general prompt "*The Eiffel Tower*". Each row is a batch of four images with the same seed.

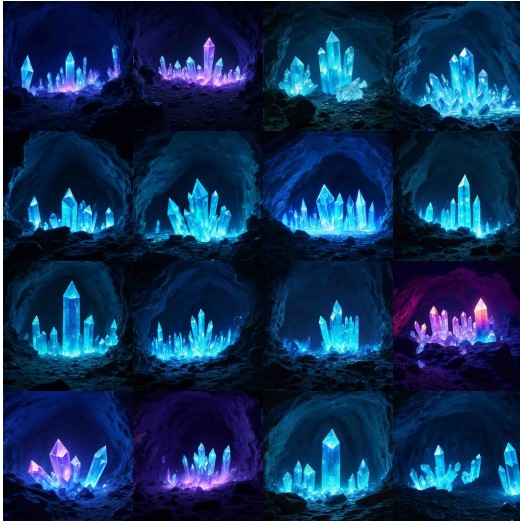
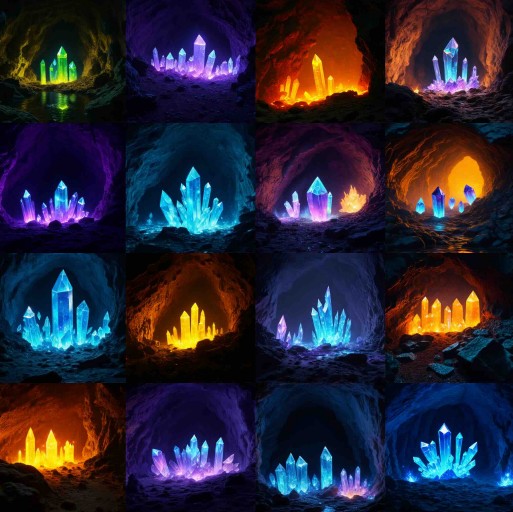

*(a)* Base model without `DivIn`   *(b)* Base model + `DivIn`

*Figure 15.* Images generated with Stable Diffusion v3.5 Medium with and without `DivIn` for general prompt "*A hidden cave with glowing crystals*". Each row is a batch of four images with the same seed.

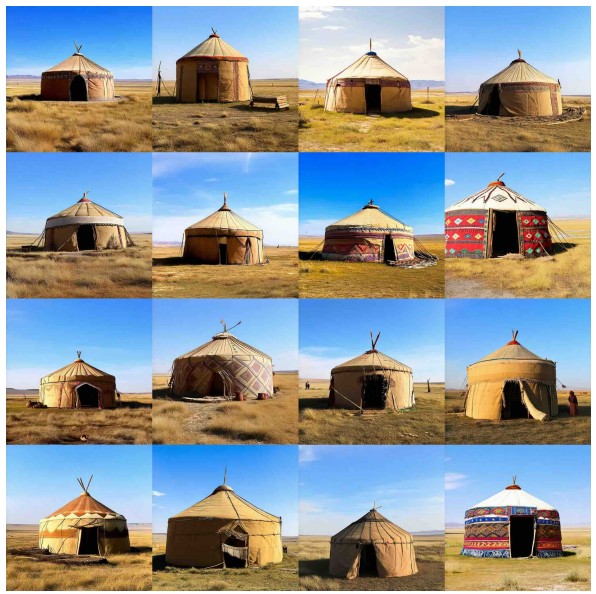

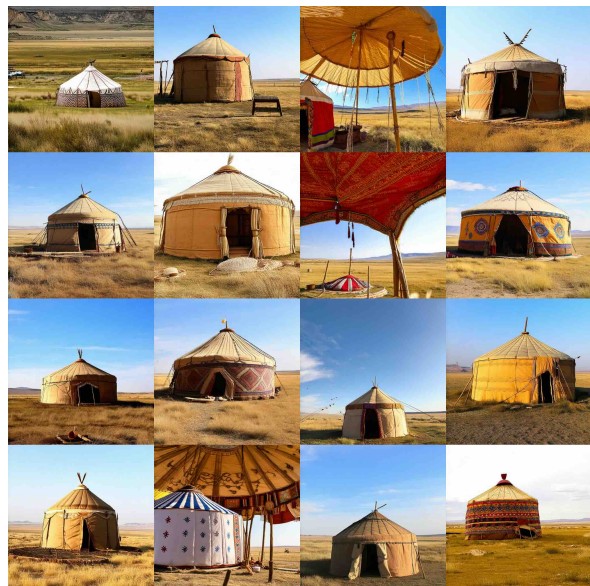

*(a)* Base model without `DivIn`

*(b)* Base model + `DivIn`

*Figure 16.* Images generated with Stable Diffusion v3.5 Medium with and without `DivIn` for general prompt "*A traditional Mongolian yurt in the steppe*". Each row is a batch of four images with the same seed.

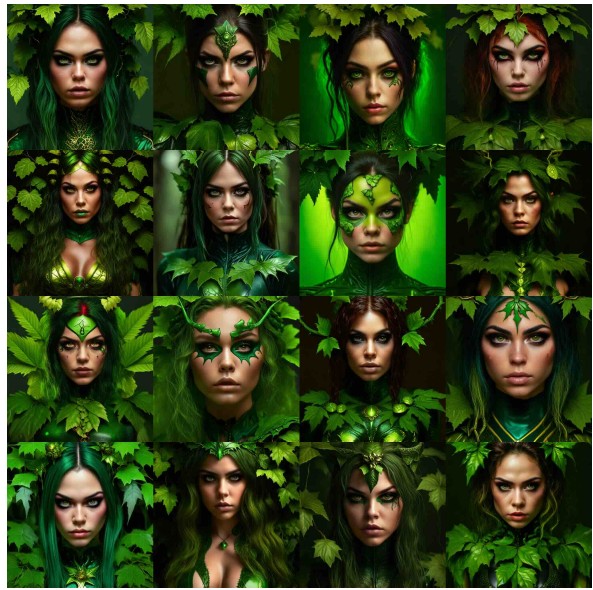

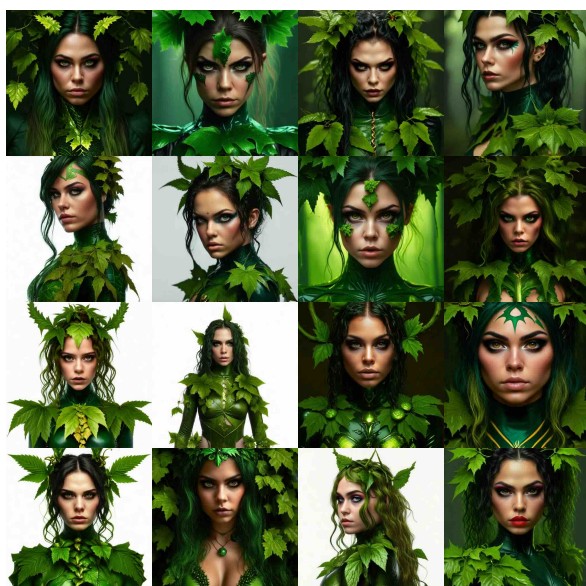

*(a)* Base model without `DivIn`

*(b)* Base model + `DivIn`

*Figure 17.* Images generated with Stable Diffusion v3.5 Medium with and without `DivIn` for general prompt "*portrait of Sporty Spice as a Poison Ivy. intricate artwork. by wlop, octane render, trending on artstation, very coherent symmetrical artwork. cinematic, hyper realism, high detail, octane render, 8k*". Each row is a batch of four images with the same seed.

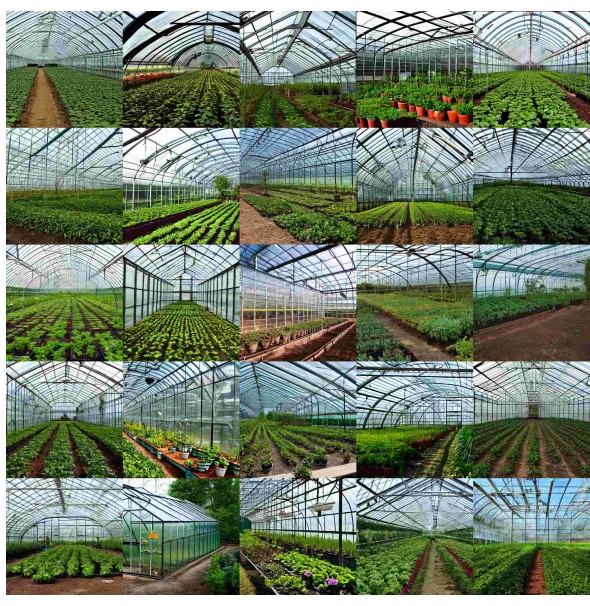
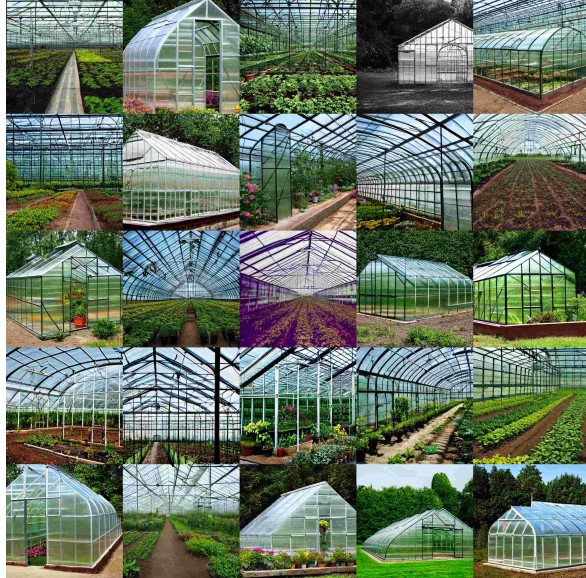

*(a)* Base model without `DivIn`

*(b)* Base model + `DivIn`

*Figure 18.* Images generated with Stable Diffusion v1.4 with and without `DivIn` for general prompt "*a photo of a greenhouse*". Each row is a batch of five images with the same seed.

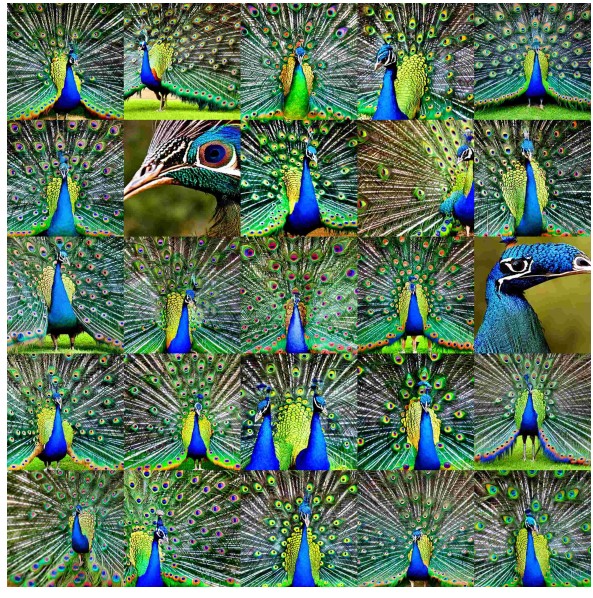
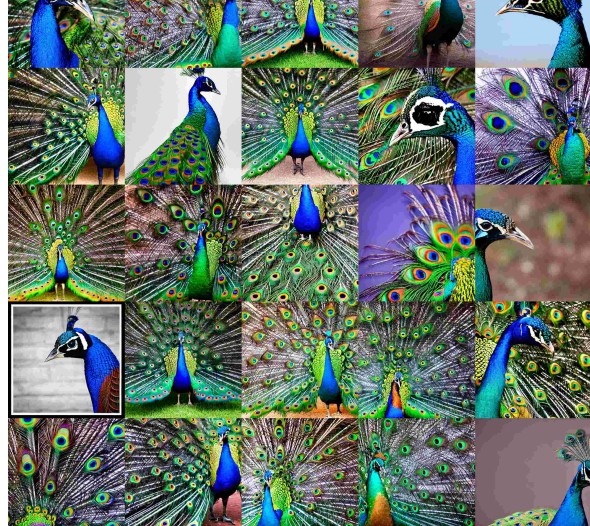

*(a)* Base model without `DivIn`

*(b)* Base model + `DivIn`

*Figure 19.* Images generated with Stable Diffusion v1.4 with and without `DivIn` for general prompt "*a photo of a peacock*". Each row is a batch of five images with the same seed.

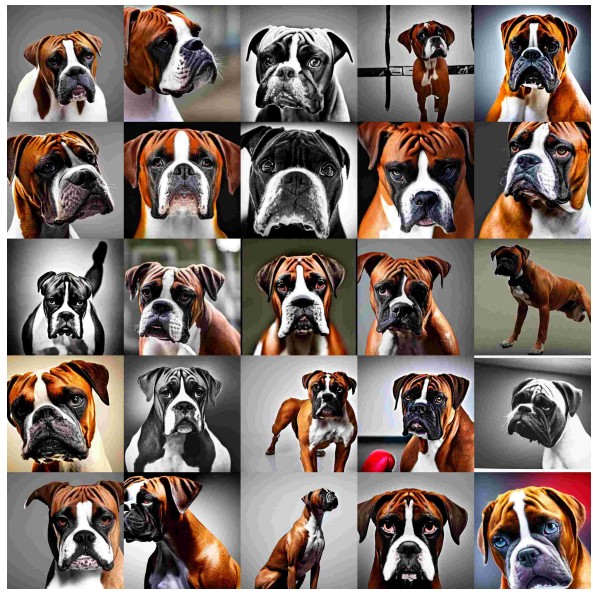

*(a)* Base model without `DivIn`

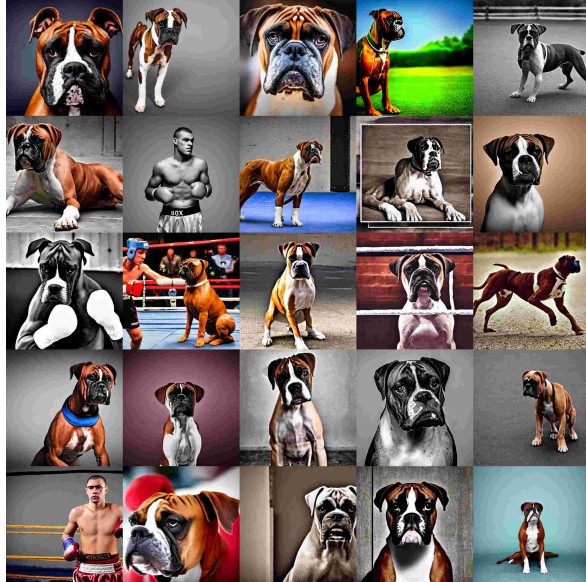

*(b)* Base model + `DivIn`

*Figure 20.* Images generated with Stable Diffusion v1.4 with and without `DivIn` for general prompt "*a photo of a boxer*". Each row is a batch of five images with the same seed.

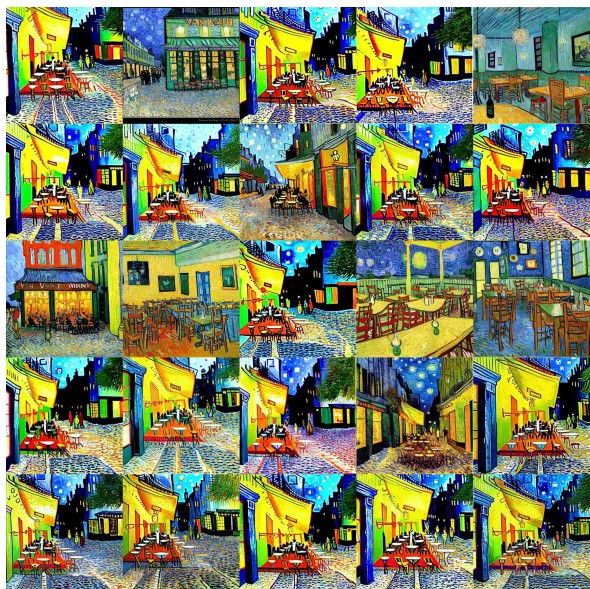

*(a)* Base model without `DivIn`

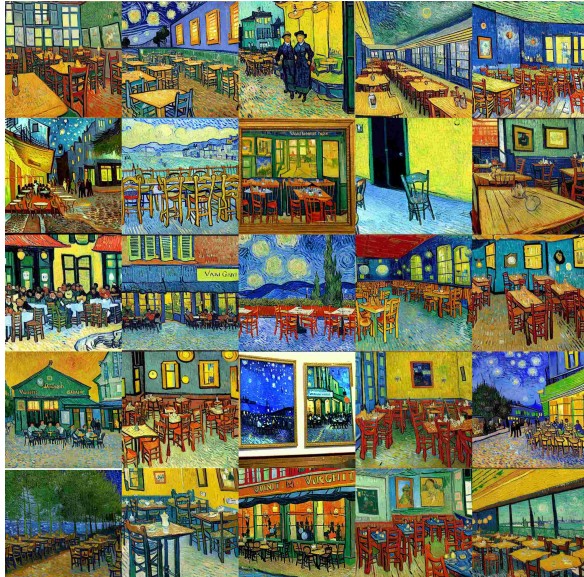

*(b)* Base model + `DivIn`

*Figure 21.* Images generated with Stable Diffusion v1.4 with and without `DivIn` for general prompt "*VAN GOGH CAFE TERASSE copy.jpg*". Each row is a batch of five images with the same seed.

