# OpenReview forum: "Initialization is Half the Battle: Generating Diverse Images from a Guidance Potential Posterior"
_ICML.cc/2026/Conference — ICML 2026 spotlight_

### Official Review · Reviewer_97PK · 2026-03-09

**Soundness:** 4
**Presentation:** 4
**Significance:** 3
**Originality:** 3
**Overall Recommendation:** 5
**Confidence:** 4

**Summary:**

This paper studies the role of initialization in the diversity of images generated by modern generative models, particularly diffusion and flow-matching models. While prior approaches mainly improve diversity by modifying the generation trajectory (e.g., through guidance strategies or trajectory-level interventions), the authors argue that the commonly used standard Gaussian initialization is agnostic to the guidance potential landscape and can bias trajectories toward dominant modes, contributing to mode collapse.

To address this issue, the paper proposes sampling the initial noise from a guidance potential posterior, which reweights the standard Gaussian prior toward regions associated with higher diversity (specifically, lower guidance potential regions). To efficiently sample from this posterior, the authors introduce Diversity-inducing Initialization (DivIn), which employs Langevin dynamics to explore the initialization landscape while maintaining compatibility with the Gaussian assumptions underlying diffusion models.

The authors also introduce a new metric of local sharpness that refines prior measurements of score landscape sharpness and enables more stable comparisons across architectures, including models beyond diffusion models. Extensive experiments on class-to-image and text-to-image generation demonstrate that DivIn improves diversity without degrading image quality. Moreover, because the approach operates at initialization rather than along the trajectory, it is orthogonal to trajectory-based diversity methods, and combining them further expands the diversity–quality Pareto frontier.

**Compliance With Llm Reviewing Policy:**

Affirmed.

**Final Justification:**

The rebuttal is helpful and addresses most of our concerns, especially those related to Q1–Q4. I keep my original score.

**Key Questions For Authors:**

Q1. Interpretation of Gaussianity in Figure 9d. In Figure 9d, the plot appears to indicate increasing Gaussianity during the initialization process. However, if Gaussianity is interpreted as the latent norm remaining close to its expected value under a standard Gaussian distribution, the trend seems to suggest the opposite. Could the authors clarify how Gaussianity is defined and how the figure should be interpreted?

Q2. Behavior of prior-free vs. noise-free settings (Table 3). In Table 3, the prior-free setting appears to lead to a smaller drop in image quality than the noise-free setting. Could the authors explain why this occurs? Intuitively, one might expect removing the prior constraint to have a stronger negative impact on quality.

Q3. Sensitivity to Langevin dynamics hyperparameters. The proposed method relies on Langevin dynamics during initialization. How sensitive is DivIn to hyperparameters such as step size, number of steps, and noise scale? Providing an ablation or discussion could help clarify the robustness of the method.

Q4. Computational overhead. Since DivIn introduces an additional sampling procedure during initialization, what is the computational overhead relative to standard diffusion sampling? A discussion of runtime impact would help assess the practicality of the approach.

**Limitations:**

Yes.

**Strengths And Weaknesses:**

S1. Soundness. I find the paper technically sound overall. The proposed formulation of sampling initial noise from a guidance potential posterior is well motivated by prior work linking initialization behavior to memorization and mode collapse in diffusion models. The use of Langevin dynamics to sample from this posterior is a reasonable and well-established approach for approximate sampling from complex distributions. Importantly, the authors explicitly address a limitation of prior work (e.g., Jeon et al.) that selects new initializations via deterministic optimization, which can violate the Gaussianity assumptions underlying diffusion models. In contrast, the stochastic formulation here maintains compatibility with the diffusion framework. The empirical evaluation is extensive and generally supports the authors’ claims that the proposed method improves diversity without sacrificing generation quality.

S2. Presentation. The paper is clearly written and well organized. The motivation, assumptions, and algorithmic steps are described in a way that makes the overall narrative easy to follow. The authors also provide clear experimental sections and detailed visualizations. Most figures and tables are well labeled and readable, and the experimental results are presented in a manner that supports the main claims. The related work section is also comprehensive and appropriately contextualizes the work within recent literature on diffusion model memorization and diversity.

S3. Significance. The paper highlights an underexplored factor affecting generative diversity—the role of initialization. Most prior approaches intervene along the generation trajectory, whereas this work focuses on improving the starting distribution itself. This perspective is interesting because it is orthogonal to many existing methods, enabling straightforward combination with trajectory-based techniques. The empirical results showing improved diversity–quality trade-offs when combining the methods suggest that this line of investigation could be practically useful.

S4. Empirical evaluation The experiments are relatively comprehensive and demonstrate improvements across multiple settings, including both class-to-image and text-to-image generation. The results consistently align with the stated goals of improving diversity while maintaining image quality.

W1. Limited originality relative to prior work
Although the paper proposes a new algorithm (DivIn), the conceptual motivation and technical framework are heavily grounded in recent work connecting initialization sharpness to memorization and diversity (e.g., Jeon et al., 2025; Han et al., 2025). As a result, the core idea—improving generative diversity by modifying initialization according to score/guidance properties—appears to be a natural extension of these earlier findings. While the stochastic Langevin formulation is a useful improvement over deterministic optimization approaches, the conceptual novelty may be somewhat incremental.

W2. Some unclear experimental details
Certain experimental results require clarification. For example, the interpretation of some plots related to Gaussianity and initialization statistics is not entirely clear, which makes it difficult to fully understand the conclusions being drawn.

W3. Minor presentation issues in specific figures
While the overall presentation quality is strong, Figure 9 in particular is somewhat confusing and may benefit from clearer explanation or improved visualization.

---

> ### Author Rebuttal · Authors · 2026-03-30
>
> We sincerely thank the reviewer for the constructive feedback. We address your concerns below and will add these clarifications and results to the revised manuscript.
>
> > W1: Originality and conceptual novelty
>
> While our geometric motivation shares common ground with recent work analyzing initialization sharpness, we respectfully clarify that our contribution represents a significant generalization and methodological improvement rather than an incremental algorithmic extension:
>
> (1) Scope: Prior works primarily investigate the narrow phenomenon of memorization triggered by memorized prompts. Our work generalizes this geometric perspective to the fundamental challenge of low generative diversity across normal prompts.
>
> (2) Methodology: prior works like SAIL employ deterministic optimization to find a single initial noise, which can be compromised to general generation. To explore diversity, we formulate initialization as sampling from a guidance potential posterior, preserving the entropy of the initialization and dispersing trajectories to discover diverse modes.
>
> (3) Application: We introduce DivIn as a universal, plug-and-play module that works orthogonally to existing trajectory-based diversity methods. This provides a fresh conceptual framework for future research: combining initialization with trajectory interventions to push the diversity-quality Pareto frontier significantly further.
>
>
> > W2 & W3: Unclear experimental details and presentation issues
>
> We clarify Figure 9 in our response to Q1 below. We would welcome any further specifics the reviewer might provide regarding other unclear details or presentation issues, ensuring we can address them comprehensively in the revision.
>
> > Q1: Clarification of Gaussianity in Figure 9
>
> We acknowledge the confusing terminology in Figure 9 and will clarify it in the revision.
> The metric labeled Gaussianity on the right y-axis is tracking the mean latent norm, $||x_T||_2$. For a standard normal distribution in $d$ dimensions, the expected $L_2$ norm is tightly concentrated around $\sqrt{d}$, which is approximately 128 for our latent space. Therefore, an ideal Gaussianity implies the norm stays close to this expected value 128. The plot should be interpreted as: (1) In Figure 9d (w/o prior), the norm grows significantly above 128, meaning the latents are drifting away from the origin and _losing_ their Gaussian prior structure; (2) In Figure 9c (w/o noise), the norm rapidly shrinks below 128. Both an exploding norm and a collapsing norm indicate a shift away from the ideal initial standard Gaussian distribution. We will rename this y-axis and add clarification in the caption to make this visualization clear.
>
> > Q2: Behavior of prior-free vs. noise-free settings
>
> This is an insightful question. The results in Table 3 used a single Langevin step, K=1, where the latent norm deviation remains relatively small. We hypothesize that this counterintuitive phenomenon occurs because diffusion models react differently to minor off-manifold initial latents. For the noise-free setting, the update becomes pure deterministic gradient descent and contracts the variance of the initial latent distribution. Diffusion models are extremely sensitive to this variance contraction, causing a slightly higher FID. In contrast, a prior-free update temporarily preserves the distribution entropy because of the injected noise term, making the model slightly more robust to it at early steps. However, when increasing DivIn steps, the unconstrained drift is amplified to catastrophic image quality degradation in both cases. In our new experimental results at K=3, image quality degrades significantly for both the noise-free (FID 17.53) and prior-free (FID 17.36) settings compared to the full DivIn (FID 15.98).
>
>
> > Q3: Sensitivity to Langevin dynamics hyperparameters
>
> For the number of steps ($K$), we provided an ablation in the left table of Figure 8 demonstrating that DivIn remains effective for $K \in \{1, 2, 3, 10\}$. We also extend this ablation to $K=20,30$ in our response to reviewer FZ5Y (W2). For step size ($\eta$) and the noise scale ($\sqrt{2\eta}$), we conducted an additional ablation study on SDv1.4, shown in the table below. Increasing the step size achieves higher Recall by exploring further into the low-potential landscape, with a trade-off in Precision. We will include this ablation in the revision.
>
> | step size ($\eta$) | noise scale ($\sqrt{2\eta}$) | Recall ↑ | Precision ↑ |
> | ------------------ | ---------------------------- | -------- | ----------- |
> | 0.01               | 0.141                        | 0.514    | 0.838       |
> | 0.05               | 0.316                        | 0.569    | 0.825       |
> | 0.1                | 0.447                        | 0.605    | 0.802       |
>
> > Q4: Computational overhead
>
> DivIn introduces ~3% additional computational cost relative to standard diffusion sampling. Please refer to our response to reviewer C7HB (W3) for details.

---

> > ### Author Rebuttal · Reviewer_97PK · 2026-04-03
> >
> > The rebuttal is helpful and addresses most of our concerns, especially those related to Q1–Q4. I keep my original score.

---

> > > ### Author Response · Authors · 2026-04-03
> > >
> > > Dear Reviewer 97PK,
> > >
> > > Thank you for your positive feedback and your acknowledgement of our rebuttal. We really appreciate your insightful comments and suggestions. We will incorporate all the discussed clarifications and additional ablations into the revised manuscript.

---

### Official Review · Reviewer_C7HB · 2026-03-09

**Soundness:** 3
**Presentation:** 3
**Significance:** 3
**Originality:** 3
**Overall Recommendation:** 4
**Confidence:** 3

**Summary:**

The paper argues that mode collapse in diffusion/flow-matching generation is often seeded at the very start: standard Gaussian noise initialization is blind to the model’s guidance potential landscape, so many trajectories begin in regions that funnel into dominant modes. It reframes initialization as sampling the starting noise from a guidance-potential posterior that reweights the usual prior toward diversity-promoting regions. To make this practical, it proposes Diversity-inducing Initialization (DivIn), which uses Langevin dynamics at inference time to search the initialization space, pushing initial noise away from collapse-prone areas while keeping it on the valid data manifold. DivIn is plug-and-play for both diffusion and flow matching, improves diversity in class-to-image and text-to-image experiments, and is complementary to trajectory-level diversity methods, with combinations yielding a better diversity–quality Pareto tradeoff than either alone.

**Compliance With Llm Reviewing Policy:**

Affirmed.

**Final Justification:**

Overall, I appreciate the authors’ efforts in clarifying parts of the paper during the rebuttal and providing additional context for their design choices. The work presents an interesting perspective on the role of initialization in flow/diffusion models and proposes a practical method to improve sample diversity at inference time.

**Soundness.** The method appears technically reasonable, and the rebuttal provides further justification for both the motivation and empirical results.

**Presentation.** The paper is clear and well structured. The rebuttal improves the explanations of the motivation and computational cost. However, some probabilistic formulations could still be described more precisely to avoid ambiguity.

**Significance.** The rebuttal better highlights the importance and practicality of inference-time methods. While this perspective is valid, the overall impact still appears somewhat limited due to the relatively modest empirical gains.

**Originality.** The work combines existing ideas in a reasonable and creative way. The rebuttal further clarifies its complementary role with respect to prior methods, which strengthens the perceived novelty.


The author has addressed most of my concerns. Given the valuable insights in this paper, I’m inclined to accept it.

**Key Questions For Authors:**

1. Will the authors release the code?
2. Please address the concerns raised in Weaknesses (1)–(3).

**Limitations:**

yes

**Strengths And Weaknesses:**

**Strengths**

1. This work presents an interesting observation: the initialization of flow/diffusion models can substantially affect the diversity of generated samples.
2. It proposes an effective solution that uses Langevin dynamics to search the initialization space, steering initial noise away from collapse-prone regions while keeping it anchored to the valid data manifold.
3. The paper is clearly written and well presented.

**Weaknesses**

1. The core motivation relies on the claim that initialization significantly impacts diversity. However, this is mainly discussed and quantified in the motivation section. Stronger experimental evidence on this phenomenon, inclding more model backbones, thorough quantitative analysis, should be necessary to support the claim. In particular, a correlation coefficient of 0.4 is too small and does not seem very compelling.

2. Improving diversity purely at inference time may not be a highly impactful problem (this is just my subjective view). The paper would benefit from a clearer discussion in the introduction of why this setting is important and practically relevant.

3. Computational overhead, including both training and inference, should be discussed more explicitly.

4. Based on Table 1, the empirical gains appear limited.

---

> ### Author Rebuttal · Authors · 2026-03-30
>
> We sincerely thank the reviewer for the constructive feedback. We address your concerns below and will add these clarifications to the revised manuscript.
>
> > W1: Experimental evidence on the core motivation
>
> We appreciate the opportunity to clarify the significance of our motivation. In the highly stochastic, high-dimensional space of image generation, a Spearman correlation of -0.4 with a p-value less than 0.001 over 1,000 prompts represents a statistically significant underlying trend. As we note in our manuscript, initialization is not the only factor influencing the final generation's diversity, as the subsequent stochastic sampling trajectories also affect the outcome. Because of this, isolating the effect from the initialization step alone is non-trivial. However, the consistent observation on correlation between potential and diversity in Fig. 4 and Fig. 11 validates that the initialization is a critical and often overlooked bottleneck in improving generative diversity.
>
> The most compelling evidence for our motivation is the consistent performance gains achieved by our proposed DivIn, as quantitatively demonstrated in Table 1, Table 2, and Figure 7. By solely reweighting the initialization distribution toward low-potential regions, DivIn achieves significant diversity improvements across both Diffusion (Stable Diffusion v1.4, Table 1) and Flow Matching (Stable Diffusion 3.5 Medium, Table 2) backbones. These results also suggest that DivIn successfully captures a unique source of diversity that trajectory-based methods miss, providing concrete empirical evidence to support our geometric motivation.
>
>
> > W2: Impact of inference-time method
>
> While we respect the reviewer's perspective, we strongly emphasize that inference-time interventions are practically and theoretically vital for the community. Training or fine-tuning large-scale foundational models such as SD-v3.5 requires massive computational resources that are inaccessible to most users. Our proposed DivIn provides a universal, plug-and-play strategy that requires zero retraining or fine-tuning. More importantly, because DivIn focuses exclusively on modifying the initial noise distribution, it is fundamentally orthogonal to existing inference-time sampling-based diversity methods (such as Particle Guidance, CADS, and Interval Guidance). Our experiments demonstrate that DivIn can be combined with these trajectory-based interventions to push the diversity-quality Pareto frontier significantly outward. This provides a distinct source of diversity that complements existing methods. We will add a clearer discussion in the introduction of the revised manuscript.
>
> > W3: Computational overhead
>
> The original manuscript analyzes the results of computational overhead in Section 5.3 (line 414) and Fig. 8. We will explicitly add a separate paragraph to discuss the computational cost in the revision. Firstly, as DivIn is a training-free method, it incurs zero training overhead. Secondly, during inference, the overhead is minimal and highly controllable. As detailed in the left table in Fig. 8, DivIn with $K=1$ requires only one additional forward and backward pass at the initial timestep, which only introduces about 3% additional wall-clock time per image, increasing the generation time from 0.754 seconds to 0.779 seconds. This makes DivIn computationally superior and significantly more stable than existing deterministic initialization baselines like SAIL, which rely on expensive second-order approximations.
>
> > W4: Limited empirical gains
>
> In the context of generative modeling, empirical gains are especially evaluated through the lens of the well-known diversity-quality trade-off. As plotted in Figure 7, the significance of our proposed DivIn's performance is that it pushes the entire diversity-quality Pareto frontier outward. In Table 1, DivIn increases the Base Model's Recall by roughly 13% (from 0.503 to 0.569) and the Vendi Score by 10% (from 4.265 to 4.688), while simultaneously improving the FID score from 16.696 to 16.158. Furthermore, combining DivIn with trajectory-based methods continues to yield notable improvements on the base performance of those methods (e.g., increasing Vendi scores from 4.395 to 4.491 for PG, and from 4.384 to 4.548 for CADS). DivIn achieving consistent improvements in both diversity and image fidelity across different backbones and datasets is a highly non-trivial enhancement of the model's capabilities.
>
> > Q1: Code release
>
> Yes, to ensure full reproducibility of our results and to support the community, we will open-source the complete codebase upon publication.

---

> > ### Author Rebuttal · Reviewer_C7HB · 2026-04-01
> >
> > The author has addressed most of my concerns. Given the valuable insights in this paper, I’m inclined to accept it.

---

> > > ### Author Response · Authors · 2026-04-01
> > >
> > > Dear Reviewer C7HB,
> > >
> > > Thank you for your positive feedback and for acknowledging our rebuttal. We appreciate your insightful comments and suggestions. We will incorporate these clarifications and improvements in our revision.

---

### Official Review · Reviewer_KKWz · 2026-03-13

**Soundness:** 2
**Presentation:** 4
**Significance:** 3
**Originality:** 2
**Overall Recommendation:** 5
**Confidence:** 4

**Summary:**

The paper tackles the problem of enhancing the sample diversity of diffusion models at inference time, in a training-free manner. The authors propose to optimizing the initial Gaussian prior commonly used in diffusion models, by first characterizing a perturbed distribution that prefers diversity, and then sampling from that distribution using Langevin dynamics. The experimental results demonstrate that the proposed approach notably improves diversity and creativity of the model. Further, as an additional benefit, the proposed noise optimization method can be used in conjunction with alternative diverse sampling strategies.

**Compliance With Llm Reviewing Policy:**

Affirmed.

**Final Justification:**

I am not fully satisfied with the response to W1, but I do believe the paper should be accepted. The rebuttal did address some of my other concerns, as to justifying why a particular formulation may be "convenient" despite several other equivalent ways to do the method. As such, I will raise my score from weak accept to accept.

**Key Questions For Authors:**

**Q1** Please see **W1**. I would like to ask the authors about the design of the toy experiment.

**Q2** Could the authors explain the settings for Fig. 4?
- From my understanding, each point in the figure represents a single prompt.
    - If so, what is the conditional guidance potential of a single prompt? Is it an average across a certain number of random trials?
    - On the Y-axis, it appears the vendi score ranges from 1-9. The vendi score is the rank approximation of a kernel, but it is not clear how big this kernel is (i.e. batch size / num images per prompt).

**Limitations:**

The paper does not appear to discuss limitations as an independent section. The notable theoretical limitation of the method is that, it is applicable for diverse sampling in conditional models only. Diverse sampling in diffusion can be desirable even in unconditional models, when the data distribution itself is highly skewed and some regions are rare. (For instance, I believe the PG baseline can do diverse sampling independently of conditions). However, I believe this is not a strong limitation; it is also present in some previous works, like CADS.

**Strengths And Weaknesses:**

### Strengths:

**S1.** From a high level perspective, the paper is generally well-written and well-motivated. Improving the diversity of diffusion-like models at inference time is a problem of interest to the field.

**S2.** Some previous work in the field implicitly rely on the iterative sampling process of diffusion models (e.g. Sadat et al., Corso et al.) via *trajectory optimization*. As recent advancements in diffusion models shift towards single (or few) step sampling, work on prior optimization, such as this paper, may prove to be useful.

**S3.** The quantitative/qualitative results in text-to-image generation generally convincing. Some of the qualitative results convincingly demonstrate the utility of the method in rare mode exploration, like Fig. 15, 19, 20.

---
---

### Weaknesses:
**W1.** I am of the opinion that the toy experiment (Fig. 3, Appendix B) misrepresents the underlying problem. I will attempt to explain why in the following.

- The authors first define a *conditional* data distribution in the form of a mixture of Gaussians, where a single central mode has higher weight. Next, the authors then define an "unconditional distribution" which shares identical modes, except the modes are uniformly weighted.
- The above choices are highly unusual. Unconditional does not necessarily mean *uniform* or balanced. In fact, it is quite possible that the unconditional distribution is an imbalanced mixture, while the conditional distribution is a uniform one (or both could be skewed, in different ways).
- Further, recall that for the conditional distribution $p(x | c)$, marginalizing over the conditions should yield the unconditional distribution, i.e. $p(x) = \sum_{c} p(x \mid c)p(c)$. In the given experiment, there is no clearly defined condition variable $c$, and no way to marginalize the conditional distributions to construct the unconditional one.
- In summary, the authors have treated two independent and unrelated distributions as “conditional” and “unconditional” respectively. In truth, they are both simply two distinct unconditional distributions, weakly related by sharing the same means (another unlikely scenario in practical data).

**W2.** I am not fully convinced by some of the justification behind Eqn (5). The authors state in L168-177 that the formulation (a) yields a metric invariant to the noise schedule, (b) unifies diffusion and flow backbones, and (c) somehow enables more robust comparison in the data manifold.

- In eqn (10), the authors show that the metric is exactly equal to the score sharpness metric (i.e. *Wen’s metric* in Jeon et al.) multiplied by a time-dependent scaling factor. As DivIn always operates at $x_T$, the scaling is always constant, i.e. the score sharpness is invariant to the noise schedule.
- I suspect using Eqn (10) would yield highly similar results as Eqn (5), even if the scaling factor is ommitted (i.e. it can just be absorbed into the temperature parameter $\tau$).
- In fact, is possible to directly extend the metric for flow matching models. The authors show in Eqn (17) that the velocity and score are proportional. The exact equality can be obtained from Lemma 1 of Zheng et al. 2023, which would give
$\\| v_\theta(x) - v_\\theta(x | c) \\| = b_t \\|s_\theta(x) - s_\theta(x | c) \\|$.
- In summary, I do not see any benefit of using Tweedie’s formula, at least theoretically, instead of using $\epsilon$ or velocity difference. An ablation could be useful here.

**W3.** The comparison done in Sec. 5.3. is not very meaningful. It is generally quite well known that the prior or initial noise in diffusion models should remain close to the isotropic Gaussian (Samuel et al. 2023), and what happens when the norm shrinks or grows w.r.t. $\sqrt{d}$. I agree with the authors that this is a weakness in the preceding work (SAIL), but one that may be handled by some simple fixes (perhaps by using projected gradient descent in SAIL updates). In general, since DivIn is also optimizing the initial latent, it may also benefit from adding a projection step after every Langevin update.


Depending on the author responses to the weaknesses, I am willing to re-evaluate my score by 1 point.

---
---
- Zheng et al. 2023, Guided Flows for Generative Modeling and Decision Making.
- Samuel et al. 2023, Norm-guided latent space exploration for text-to-image generation.

---

> ### Author Rebuttal · Authors · 2026-03-30
>
> We sincerely thank the reviewer for the insightful questions. Below, we address these concerns in detail.
>
> > W1&Q1: Design of the toy experiment
>
> We completely agree that real-world conditional and unconditional distributions are highly complex, and we acknowledge that treating them independently sacrifices mathematical rigor.
> Our design choice was not intended to be a strict global statistical simulation, but rather an idealized **local approximation** built for **controlled geometric visualization**. In the neighborhood of a target conditional mode, the unconditional landscape can be generally much flatter than the sharp basin of attraction formed by the conditional distribution. By modeling the unconditional distribution $p(x)$ as uniform, we establish a control variable to approximate this relative and localized flatness. This decoupling allows us to purely **isolate the impact of the conditioning signal**, visually demonstrating how the sharp geometry of $p(x|c)$ alone induces mode collapse. In the revision, we will clarify the toy experiment setting and explicitly acknowledge this theoretical relaxation.
>
>
> > W2: Justification for the Tweedie Formulation
>
> We entirely agree that for a fixed timestep, the scaling factor is constant, and for flow matching models, exact equality to the velocity difference holds. We will include this theoretical connection and cite Zheng et al. (2023) in the revision.
>
> Our formulation of $U(x_T, c)$ in the estimated data space is driven by practical considerations to ensure our method remains robust and plug-and-play. Firstly, projecting to $\hat{x}_0$ unifies the potential computation across diffusion and flow matching architectures into a shared Euclidean pixel space.
> Secondly, in standard deployment pipelines, the initial continuous timestep $t=T-1$ varies depending on the chosen noise scheduler and total number of inference steps (e.g., mapping to a 1000-step training schedule, a 50-step DDIM schedule starts at $t=981$, while a 30-step schedule starts at $t=958$).
> If we used the raw score difference $U = \\|\Delta s\_{\theta}(x)\\|$ and absorbed the scaling factor ($\tau' = \lambda_t \cdot \tau$), the optimal temperature $\tau'$ would become extremely brittle.
> Our ablation below demonstrates this. While both perform similarly *when perfectly tuned*, the score difference formulation (Eqn 10) requires wildly retuned temperatures (130.7 vs. 171.5) simply by changing the inference steps. Conversely, our Tweedie formulation (Eqn 5) intrinsically handles this dynamic $\lambda_t$ scaling, keeping the optimal temperature perfectly stable ($\tau = 0.6$).
>
> We will clarify this as a practical justification rather than a theoretical difference in the revision.
>
> | Inference steps | $\tau$ (Tweedie-based) | Recall ↑ | FID ↓ | $\tau'=\lambda_t\cdot\tau$ (score-based) | Recall ↑ | FID ↓ |
> | --------------- | ---------------------- | -------- | ----- | -------------------------------------- | -------- | ----- |
> | 30              | 0.6                    | 0.585    | 16.28 | 130.7                                  | 0.585    | 16.29 |
> | 50              | 0.6                    | 0.569    | 16.16 | 171.5                                  | 0.565    | 16.22 |
>
> > W3: The comparison done in Sec. 5.3.
>
> We thank the reviewer for pointing us to Samuel et al. (2023), which we will explicitly cite in our revision. The comparison in Sec. 5.3 was not intended to present the latent norm preservation as a novel finding, but to highlight the fundamental conflict between deterministic optimization and generative diversity. To investigate your hypothesis, we ran the suggested experiments using projected gradient descent on both methods. The results below show that while projection fixes SAIL's shrinking norm and slightly improves precision, it sacrifices diversity, causing the Vendi score to drop significantly (by 0.093). Because of SAIL's deterministic, mode-seeking nature, projection does not prevent the underlying collapse of the distribution volume. In contrast, applying projection to DivIn results in a negligible Vendi drop (0.009). DivIn's Langevin dynamics intrinsically respect the prior, preserving entropy to enable diverse exploration. We will include this and revise Sec. 5.3 in the revision.
>
> | Method     | Vendi ↑ | Precision ↑ |
> | ---------- | ------- | ----------- |
> | SAIL       | 4.549   | 0.825       |
> | SAIL+proj  | 4.456   | 0.828       |
> | DivIn      | 4.688   | 0.825       |
> | DivIn+proj | 4.679   | 0.835       |
>
>
> > Q2: Settings for Fig. 4
>
> We sampled 10 independent initial noises ($x_T$) for each of the 1000 distinct prompts. The x-axis represents the average potential computed across the 10 latents for a specific prompt. And the y-axis Vendi score is calculated on a kernel of 10, which aligns with the observed 1-9 range in the plot. We will clarify these settings explicitly in the figure caption.

---

> > ### Author Rebuttal · Reviewer_KKWz · 2026-04-03
> >
> > The authors have addressed all points adequately.

---

> > > ### Author Response · Authors · 2026-04-03
> > >
> > > Dear Reviewer KKWz,
> > >
> > > We appreciate your positive feedback and your acknowledgement of our rebuttal. We really appreciate your insightful comments and suggestions. We will incorporate all the discussed clarifications and additional ablations into the revised manuscript.

---

### Official Review · Reviewer_FZ5Y · 2026-03-16

**Soundness:** 3
**Presentation:** 4
**Significance:** 3
**Originality:** 3
**Overall Recommendation:** 5
**Confidence:** 5

**Summary:**

The paper tackles the task of generating diverse images from a text-to-image diffusion model using classifier-free guidance by altering the initial noise distribution. The authors propose avoiding samples from regions with high conditional guidance potential, where the probability contracts rapidly and entropy decreases quickly, leading to similar-looking samples. To achieve this, they introduce a computationally efficient proxy, $U(x_T, c)$, for this potential and use Langevin dynamics to sample initial latents from $x_T \sim \exp(-\frac{1}{2}|x_T|_2^2)\times \exp(-\tau U(x_T, c))$, followed by standard flow-based sampling. Experimentally, they evaluate the approach on text-to-image generation and demonstrate improved diversity while maintaining high image quality. They also show that the method is orthogonal to and compatible with prior diversity-enhancing trajectory-guidance approaches.

**Compliance With Llm Reviewing Policy:**

Affirmed.

**Final Justification:**

The paper is solid, and the few questions/concerns I had was fully addressed in the rebuttal. Therefore, I recommend accepting this paper.

**Key Questions For Authors:**

- Theoretically, for an optimal diffusion or flow-matching model, we would have $\hat{x_0}(x_T, c) = E_{x_0 \sim p(x \mid c)}[x_0]$ for all $x_T$, assuming $x_T \sim \mathcal{N}(0, I)$. In that case, the proxy $U(x_T, c)$ would be constant everywhere, making the Langevin dynamics updates effectively useless. How consistent is this observation with what you have seen in practice? More generally, why does DivIn seem to work so well despite the theory suggesting otherwise?

- Following the previous question, how compatible is this approach with few-step generators? Would replacing the current $\hat{x}_0$, which represents the expected $x_0$ value, with $\tilde{x}_0$, the few-step output of a distilled model, break the underlying geometric assumptions behind the potential?

- Intuitively, the proposed potential steers initial samples away from regions where entropy is reduced rapidly and many initial noises collapse to similar outputs. Is it possible that, when DivIn is run for many steps, the resulting algorithm may miss the dominant mode that was previously being covered through this collapse? For example, in the first row of Figure 6, the famous starry night scene captured three times by the base model is no longer seen on the right-hand side. If so, is there a more fundamental way to address this beyond just lowering the temperature $\tau$?

**Limitations:**

Yes.

**Strengths And Weaknesses:**

Strengths:

- The paper is very well written and easy to follow.
- The problem is well motivated and very timely. The diversity of generations from diffusion models is a uniquely challenging problem right now, and it is becoming increasingly relevant post-training on these models often leads to reduced output diversity. The proposed approach is simple, intuitive, and easy to implement.
- The experimental results are promising, with the method improving almost all metrics while remaining compatible with many prior diversity-enhancing methods.

Weaknesses:

This is a strong submission, and there are no major weaknesses to highlight. That said, there are a few minor issues:

- Figure 5 is somewhat ambiguous. What exactly are the red and blue colors visualizing? It appears that the stochastic Langevin dynamics approach completely misses or drops the left mode of the blue distribution, whereas SAIL appears to cover it. Is that a preferable outcome?
- The table in Figure 8 shows that as the number of Langevin steps increases, recall and FID increase at high step counts. Intuitively, as more Langevin steps are performed, the sampled points should more closely approximate the potential-adjusted prior. Does this FID trend continue as the number of steps increases further? Do the output images gradually degrade as as this continues?

---

> ### Author Rebuttal · Authors · 2026-03-30
>
> We sincerely thank the reviewer for the insightful questions. Below, we address your concerns and questions in detail.
>
> > W1: Clarification on Fig. 5
>
> The background colors visualize the **guidance potential landscape at the initial denoise step**, _not_ the final target data distribution. Red regions denote high-potential areas where conditional guidance exerts a strong contractive pull towards a dominant mode. Blue regions denote lower-potential areas, where the left blue region is a **sharp local minimum (a trap)** and the right blue region is a **flat minimum basin**. The red and green dots represent the **optimized initial noise latents** $x_T$, not the final generated samples $x_0$.
>
> In this figure, SAIL suffers from (1) **Volume collapse**: SAIL deterministically optimizes initial latents to collapse into a single point, destroying the initialization's entropy; (2) **Sharp traps**: SAIL gets trapped in the left sharp local minimum, from where the subsequent reverse process is highly fragile and is likely to be pulled back into the central dominant mode. Conversely, DivIn preserves volume by stochastically dispersing latents into the right flat basin for diverse generation. Thus, dropping the left region is indeed the preferable outcome.
>
> We will update the clarification in the revision.
>
> > W2: Table in Fig. 8 about Langevin steps and FID
>
> The image quality does not continuously degrade as the steps increase.  We extended the evaluation of DivIn on SD v1.4 up to 30 steps in the table below. As shown below, rather than diverging, both FID and Recall quickly reach a stable equilibrium after step 10. The observed trend reflects that the Langevin dynamics process converges to the **stationary distribution** of the target posterior from step 10 to 30. Thus, the metrics stabilize (FID around 17.0, Recall around 0.61) without further degradation. We will include this discussion in the revised version.
>
> | **Step (K)** | **10** | **20** | **30** |
> | ------------ | ------ | ------ | ------ |
> | **Recall ↑** | 0.609  | 0.617  | 0.614  |
> | **FID ↓**    | 17.02  | 17.04  | 16.99  |
>
> > Q1: Theoretical and practical landscapes of $U$
>
> We agree that  $U$ is constant at the **theoretical noise limit** ($t \to \infty$) for an *optimal* model.
> However, practical inference starts at a finite timestep $t=T-1$, eg, mapping to a 1000-step training schedule, a 50-step DDIM schedule effectively begins at $t=981$. While the initial latent $x_T$ is indeed sampled from pure Gaussian noise $\mathcal{N}(0, I)$, the **model's learned score landscape** at this finite timestep is not flat, and thus results in a non-zero gradient field. The model's conditional prediction $\hat{x}_0(x_T, c)$ diverges significantly from its unconditional prediction $\hat{x}_0(x_T, \emptyset)$. Therefore, the initialization potential $U(x_T, c)$ varies across the manifold. This is empirically proven in Figure 9(a): the mean potential decreases over iteration steps, validating a non-zero gradient field ($\nabla U \neq 0$) exists at $t=T-1$, guiding latents toward flatter regions. We will clarify this theoretical and practical distinction in the revision.
>
> > Q2: Compatibility with few-step generators
>
> Replacing the one-step estimate $\hat{x}_0$ with a distilled few-step output $\tilde{x}_0$ relaxes the strict mathematical equality to the local Hessian, but **preserves the geometric intuition**. The core mechanism of our guidance potential remains fundamentally valid by measuring the divergence between the conditional and unconditional predictions. Even evaluating $\\|\tilde{x}_0(c) - \tilde{x}_0(\emptyset)\\|$ in a distilled model, a high potential geometrically defines a rapid attraction to the specific mode. Minimizing this proxy potential guides the initialization away from these high-potential regions that are prone to mode collapse, making DivIn compatible with few-step generators.
>
> > Q3: Over-suppression of dominant modes
>
> Mathematically, DivIn samples from a modified posterior $p_{diverse} \propto \exp(-\tau U) \cdot p(x_T)$ that  **reweights probability mass** rather than eliminating the dominant mode. In our extended empirical test, with $K=10$ steps on the "Van Gogh painting" prompt, the dominant "Starry Night" still appears in 8 out of 50 generated images for DivIn, demonstrating that it is under-sampled rather than completely missed. A fundamental solution could be formulate the initialization as an explicit mixture distribution $p_{mix}(x_T | c) = (1-\lambda)\mathcal{N}(0, I) + \lambda p_{diverse}(x_T | c)$, where a fixed proportion of the initial latents is strictly drawn from the standard prior to recover the dominant modes. We will add a discussion of this mixture formulation as future work.

---

> > ### Author Rebuttal · Reviewer_FZ5Y · 2026-04-03
> >
> > I thank the authors for their response. My questions were all addressed and as such, I raise my score.

---

> > > ### Author Response · Authors · 2026-04-03
> > >
> > > Dear Reviewer FZ5Y,
> > >
> > > Thank you for your positive feedback and for acknowledging our rebuttal. We really appreciate your constructive comments and suggestions. We will incorporate all the clarifications and additional experimental results into the revised manuscript.

---

### Decision · Program_Chairs · 2026-04-30

**Decision:**

Accept (spotlight)

**Comment:**

DivIn is plug-and-play, orthogonal to trajectory-based diversity methods for diffusion models. All reviewers agreed this paper is clearly written, well organized, and technically sound, with a well-motivated insight that standard Gaussian initialization biases trajectories toward dominant modes, contributing to mode collapse. The proposed DivIn uses Langevin dynamics to sample initial noise from a guidance potential posterior, improving diversity without degrading generation quality.

The authors’ rebuttal addressed the main concerns of reviewers. Therefore, I recommend acceptance, and encourage the authors to incorporate the discussed clarifications in the camera-ready version.